# AlignIQL: Policy Alignment in Implicit Q-Learning through Constrained Optimization

## Abstract

Implicit Q-learning (IQL) serves as a strong baseline for offline RL, which never needs to evaluate actions outside of the dataset through quantile regression. However, it is unclear how to recover the implicit policy from the learned implicit Q-function and whether IQL can utilize weighted regression for policy extraction. IDQL reinterprets IQL as an actor-critic method and gets weights of implicit policy, however, this weight only holds for the optimal value function under certain critic loss functions. In this work, we introduce a different way to solve the *implicit policy-finding problem* (IPF) by formulating this problem as an optimization problem. Based on this optimization problem, we further propose two practical algorithms AlignIQL and AlignIQL-hard, which inherit the advantages of decoupling actor from critic in IQL and provide insights into why IQL can use weighted regression for policy extraction. Compared with IQL and IDQL, we find that our method keeps the simplicity of IQL and solves the implicit policy-finding problem. Experimental results on D4RL datasets show that our method achieves competitive or superior results compared with other SOTA offline RL methods. Especially in complex sparse reward tasks like AntMaze, our method outperforms IQL and IDQL by a significant margin.

## 1 Introduction

Offline Reinforcement Learning (RL), or Batch RL aims to seek an optimal policy without environmental interactions (Fujimoto et al., 2019; Levine et al., 2020). This is compelling for having the potential to transform large-scale datasets into powerful decision-making tools and avoid costly and risky online environmental interactions, which offers significant application prospects in fields such as healthcare (Nie et al., 2021; Tseng et al., 2017) and autopilot (Yurtsever et al., 2020; Rhinehart et al., 2018). Notwithstanding its promise, applying off-policy RL algorithms (Lillicrap et al., 2015; Fujimoto et al., 2018; Haarnoja et al., 2018a;b) directly into the offline context presents challenges due to out-of-distribution actions that arise when evaluating the learned policy.(Fujimoto et al., 2019; Levine et al., 2020).

Although a variety of methods based on constraint and conservative Q-learning have been proposed to address this problem, IQL (Kostrikov et al., 2021b) stands out among them since IQL avoids visiting out-of-distribution (OOD) actions and decouples the critic from the actor, which contributes to stability and hyperparameter robust. For implicit policy extraction, IQL extracts policy through advantage-weighted regression (AWR) (Nair et al., 2020; Peng et al., 2019; Peters et al., 2010). However, the general form of extracted policy is $\pi(\boldsymbol{a}|\boldsymbol{s}) \propto \mu(\boldsymbol{a}|\boldsymbol{s})w(\boldsymbol{s}, \boldsymbol{a})$, where $\mu(\boldsymbol{a}|\boldsymbol{s})$ is the behavior policy. The AWR's weight used by IQL is obtained from the constrained policy search, which does not guarantee that it is the policy the learned IQL's value function is actually evaluating (Hansen-Estruch et al., 2023).

To solve this problem, IDQL (Hansen-Estruch et al., 2023) reinterprets IQL as an actor-critic method and derives the implicit optimal policy weights. Nevertheless, this optimal weight hinges on the assumption that the optimal value function can be learned under certain critic loss functions. It remains unclear whether using AWR to extract policies for IQL is feasible, and how to extract policies from arbitrary critic loss function, not just the expectile loss. Recently, this issue has become more important since 1) many recent offline RL (Chen et al., 2023) and safe RL methods (Zheng et al., 2023) use IQL to learn the Q-function; 2) IQL's performance is significantly affected by the choice

of a policy extraction algorithm (Tarasov et al., 2024). Addressing these issues can lead to a better understanding of IQL-style methods' bottlenecks, thereby promoting the development of offline RL.

In this paper, we address the above issues by formulating the implicit policy-finding problem as an optimization problem, where the objective function is a generalized form of behavior regularizers and the constraint is policy alignment. Policy alignment ensures the extracted policy is the policy implied in the Q-function. By solving this optimization problem, we can get a closed-form solution, which can be expressed by imposing weight on the behavior policy. The weight consists of a value function, an action-value function, and multipliers, indicating that using AWR to extract IQL policies is feasible only when the certain multiplier is less than 0, and this conclusion can be generalized to any value function loss. Furthermore, our work also explains how the implicit policy in IQL-style methods addresses OOD actions from the perspective of behavior regularizers.

Based on the optimization problem, we further propose two algorithms, AlignIQL-hard and AlignIQL. Both inherit the characteristics of IQL, $i.e.$ the decoupling of actor and critic training. AlignIQL-hard can theoretically achieve a globally optimal solution, but it is more vulnerable to hyperparameter choices than AlignIQL. AlignIQL relaxes the policy alignment constraint and performs better in complex tasks like sparse rewards tasks, but it does not guarantee convergence to the global optimum.

Recently, Diffusion models (Sohl-Dickstein et al., 2015; Ho et al., 2020; Song & Ermon, 2019) have been widely used in Offline RL, since behavior policy is often complex and potentially multimodal, the unimodal Gaussian policy used in IQL is unlikely to accurately approximate the complex behavior policy (Wang et al., 2022; Hansen-Estruch et al., 2023; Chen et al., 2022; He et al., 2024), which in turn affects the implicit policy extraction. Our method can also be easily combined with diffusion models. We just need to resample the actions generated by the diffusion-parameterized behavior model according to the weights $w(s, a)$ of our method. We verify the effectiveness of our method on D4RL datasets and Atari datasets and demonstrate its state-of-the-art (SOTA) performance on sparse reward datasets like AntMaze tasks in which the learning of the critic network is hard and unstable. We also show that, compared to IDQL, AlignIQL is more robust to hyperparameters and achieves more stable training.

To summarize, our main contributions are as follows:

- We propose the policy-finding problem, where the policy alignment term is added as a constraint. By solving this problem, we provide insights into why and when IQL can use weighted regression for policy extraction, and in turn, make it better to understand the bottlenecks of the IQL-style algorithms.

- We demonstrate that there is no price to achieving policy alignment in IQL-style methods, all we need is to modify the importance weights of the extracted policy. These results can be generalized to any generalized value loss function, which greatly extends the theoretical results of IDQL.

- We introduce two IQL-style algorithms: AlignIQL-hard and AlignIQL. AlignIQL-hard can theoretically achieve a globally optimal solution. AlignIQL obtains SOTA performance on D4RL AntMaze tasks and show robustness on sparse reward tasks.

## 2 RELATED WORK

**Offline RL**. Offline RL algorithms need to avoid OOD actions. Previous methods to mitigate this issue under the model-free offline RL setting generally fall into three categories: 1) value function-based approaches, which implement pessimistic value estimation by assigning low values to out-of-distribution actions (Kumar et al., 2020; Fujimoto et al., 2019), or implicit TD backups (Kostrikov et al., 2021b; Ma et al., 2021) to avoid the use of out-of-distribution actions 2) sequential modeling approaches, which casts offline RL as a sequence generation task with return guidance (Chen et al., 2021; Janner et al., 2022; Liang et al., 2023; Ajay et al., 2022), and 3) constrained policy search (CPS) approaches, which regularizes the discrepancy between the learned policy and behavior policy (Peters et al., 2010; Peng et al., 2019; Nair et al., 2020).

**Implicit Q-learning**. Recently, *implicit Q-learning* (Kostrikov et al., 2021b) has attracted interest due to its stable training and simplicity. Many offline RL methods (Chen et al., 2023; Zheng

et al., 2023; Hansen-Estruch et al., 2023) use IQL-style expectile regression to learn Q-function and realize the advantage of decoupling the training of actor and critic. While IQL achieves superior performance, several issues remain unsolved. SQL (Xu et al., 2023) reinterprets IQL in the Implicit Value Regularization (IVR) framework and provides insights about why in practice a large $\tau$ may give a worse result in IQL. However, there is another important open question about IQL, that is, what policy the learned value function is evaluating. IDQL (Hansen-Estruch et al., 2023) solves this by reinterpreting the IQL as an actor-critic method and getting the corresponding implicit policy for the (generalized) IQL loss function. However, the corresponding implicit policy in IDQL only holds for optimal value function under certain critic loss functions.

The closest work to ours is IDQL (Hansen-Estruch et al., 2023), which derives the implicit policy for optimal value function under different critic loss functions. Our method is related, but features with AlignIQL can be applied to arbitrary sub-optimal value functions and arbitrary critic loss functions. More importantly, our method explains when and why IQL can use AWR for policy extraction while providing theoretical insights for IQL and other RL paradigms that use Q-values to guide sampling.

## 3 BACKGROUND

**Offline RL.** Consider a Markov decision process (MDP): $M = \{\mathcal{S}, \mathcal{A}, P, R, \gamma, d_0\}$, with state space $S$, action space $\mathcal{A}$, environment dynamics $\mathcal{P}(\boldsymbol{s}'|\boldsymbol{s}, \boldsymbol{a}) : S \times S \times \mathcal{A} \to [0, 1]$, reward function $R : S \times \mathcal{A} \to \mathbb{R}$, discount factor $\gamma \in [0, 1)$, policy $\pi(\boldsymbol{a}|\boldsymbol{s}) : \mathcal{S} \times \mathcal{A} \to [0, 1]$, and initial state distribution $d_0$. The action-value or Q-value of policy $\pi$ is defined as $Q^\pi(\boldsymbol{s}_t, \boldsymbol{a}_t) = \mathbb{E}_{\boldsymbol{a}_{t+1}, \boldsymbol{a}_{t+2}, \dots \sim \pi} \left[ \sum_{j=0}^\infty \gamma^j r(\boldsymbol{s}_{t+j}, \boldsymbol{a}_{t+j}) \right]$. The goal of RL is to get a policy to maximize the cumulative discounted reward $J(\phi) = \int_{\mathcal{S}} d_0(\boldsymbol{s}) Q^\pi(\boldsymbol{s}, \boldsymbol{a}) d\boldsymbol{s}$. $d^\pi(\boldsymbol{s}) = \sum_{t=0}^\infty \gamma^t p_\pi(\boldsymbol{s}_t = \boldsymbol{s})$ is the state visitation distribution induced by policy $\pi$ (Sutton & Barto, 2018; Peng et al., 2019), and $p_\pi(\boldsymbol{s}_t = \boldsymbol{s})$ is the likelihood of the policy being in state $\boldsymbol{s}$ after following $\pi$ for $t$ timesteps. In offline setting (Fujimoto et al., 2019), environmental interaction is not allowed, and a static dataset $\mathcal{D} \triangleq \{(\mathcal{S}, \mathcal{A}, R, \mathcal{S}', \text{done})\}$ is used to learn a policy.

**Advantage Weighted Regression (AWR).** Prior works (Peters et al., 2010; Peng et al., 2019) formulate offline RL as a constrained policy search (CPS) problem with the following form:

$$\pi^* = \arg\max_\pi J(\pi) = \arg\max_\pi \int_{\mathcal{S}} d_0(\boldsymbol{s}) \int_{\mathcal{A}} \pi(\boldsymbol{a}|\boldsymbol{s}) Q^\pi(\boldsymbol{s}, \boldsymbol{a}) d\boldsymbol{a} d\boldsymbol{s}$$
$$s.t. \quad D_{\text{KL}}(\mu(\cdot|\boldsymbol{s})\|\pi(\cdot|\boldsymbol{s})) \leq \epsilon, \quad \forall \boldsymbol{s} \tag{1}$$
$$\int_{\boldsymbol{a}} \pi(\boldsymbol{a}|\boldsymbol{s}) d\boldsymbol{a} = 1, \quad \forall \boldsymbol{s},$$

Previous works (Peters et al., 2010; Peng et al., 2019; Nair et al., 2020) solve Equation 1 through KKT conditions and get the optimal policy $\pi^*$ as:

$$\pi^*(\boldsymbol{a}|\boldsymbol{s}) = \frac{1}{Z(\boldsymbol{s})} \mu(\boldsymbol{a}|\boldsymbol{s}) \exp\left(\alpha Q_\theta(\boldsymbol{s}, \boldsymbol{a})\right), \tag{2}$$

where $Z(\boldsymbol{s})$ is the partition function, $\alpha \geq 0$ is a Lagrange multiplier, and $Q_\theta$ is a learned Q-function of the current policy $\pi$. Intuitively we can use Equation 2 to optimize policy $\pi$. However, the behavior policy may be very diverse and hard to model. To avoid modeling the behavior policy, prior works (Peng et al., 2019; Wang et al., 2020; Chen et al., 2020) optimize $\pi^*$ through a parameterized policy $\pi_\phi$, known as AWR:

$$\arg\min_\phi \mathbb{E}_{\boldsymbol{s} \sim \mathcal{D}^\mu} \left[ D_{\text{KL}} \left( \pi^*(\cdot|\boldsymbol{s}) \| \pi_\phi(\cdot|\boldsymbol{s}) \right) \right] \tag{3}$$

$$= \arg\max_\phi \mathbb{E}_{(\boldsymbol{s}, \boldsymbol{a}) \sim \mathcal{D}^\mu} \left[ \frac{1}{Z(\boldsymbol{s})} \log \pi_\phi(\boldsymbol{a}|\boldsymbol{s}) \exp\left(\alpha Q_\theta(\boldsymbol{s}, \boldsymbol{a})\right) \right].$$

where $\exp(\alpha Q_\theta(\boldsymbol{s}, \boldsymbol{a}))$ being the regression weights.

**Implicit Q-learning (IQL).** To avoid OOD actions in offline RL, IQL (Kostrikov et al., 2021a) uses the state conditional upper expectile of action-value function $Q(\boldsymbol{s}, \boldsymbol{a})$ to estimate the value function

$V(s)$, which avoid directly querying a Q-function with unseen action. For a parameterized critic $Q_\theta(s, a)$, target critic $Q_{\hat{\theta}}(s, a)$, and value network $V_\psi(s)$ the value objective is learned by

$$\mathcal{L}_V(\psi) = \mathbb{E}_{(s,a) \sim \mathcal{D}}[L_2^\tau(Q_{\hat{\theta}}(s, a) - V_\psi(s))]$$
$$\text{where} \quad L_2^\tau(u) = |\tau - \mathbb{1}(u < 0)|u^2, \tag{4}$$

where $\mathbb{1}$ is the indicator function. Then, the Q-function is learned by minimizing the MSE loss

$$\mathcal{L}_Q(\theta) = \mathbb{E}_{(s,a,s') \sim \mathcal{D}}[(r(s, a) + \gamma V_\psi(s') - Q_\theta(s, a))^2]. \tag{5}$$

Note that, in IQL, the policy is not explicitly represented, it is implicit in the learned value function. For policy extraction, IQL uses Equation 3 in AWR (Peters et al., 2010; Peng et al., 2019; Nair et al., 2020), which trains the policy through weighted regression by minimizing

$$\mathcal{L}_\pi(\phi) = \mathbb{E}_{(s,a) \sim \mathcal{D}}[-\exp(\alpha(Q_{\hat{\theta}}(s, a) - V_\psi(s))) \log \pi_\phi(a|s)]. \tag{6}$$

However, it is still unclear whether AWR can be used to extract policies for IQL. Answering this question can help us better understand the bottlenecks of IQL-style methods.

## 4 IMPLICIT POLICY-FINDING PROBLEM

Before drawing our method, we first introduce the form of the Implicit Policy-finding Problem. Firstly, we introduce Definition 4.1, which defines what a policy implied by a value function is (*i.e.* Policy alignment).

**Definition 4.1.** We refer to a policy as one implied by the value function $Q(s, a), V(s)$, when

$$Q(s, a) - r(s, a) - \gamma \mathbb{E}_{s' \sim p(s'|s,a), a' \sim \pi(a'|s')}[Q(s', a')] = 0. \tag{7}$$

$$\mathbb{E}_{a \sim \pi(a|s)}[Q(s, a)] = V(s), \tag{8}$$

Definition 4.1 is derived from IDQL (Hansen-Estruch et al., 2023) and the conventional definition of the value function in actor-critic methods. Note that in IQL, the $Q$-function is updated by minimizing Equation 5, which implies if we can ensure Equation 8, Equation 7 can be derived by substituting Equation 8 back in Equation 5. So in the following sections, we eliminate Equation 7 and use Equation 8 as policy alignment constraint.

It is known that the offline RL problem can be solved by constrained policy search (CPS) problem (aka AWR) (Nair et al., 2020; Peng et al., 2019; Peters et al., 2010), where a policy is sought to maximize cumulative rewards under the constraint of policy divergence from the behavior policy. Inspired by CPS, we formulate the *implicit policy-finding problem* (IPF) as a constrained optimization problem, where a policy is sought to minimize policy divergence from the behavior policy under policy alignment

$$\min_\pi \quad \mathbb{E}_{s \sim d^\pi(s), a \sim \pi(a|s)}\left[f\left(\frac{\pi(a|s)}{\mu(a|s)}\right)\right]$$
$$s.t. \quad \pi(a|s) \geq 0, \quad \forall s, \forall a$$
$$\int_a \pi(a|s)da = 1, \quad \forall s \tag{IPF}$$
$$\mathbb{E}_{a \sim \pi(a|s)}[Q(s, a)] - V(s) = 0, \quad \forall s,$$

where $V(s), Q(s, a)$ is the learned value function, which does not have to be the optimal value function. $f(\cdot)$ is a regularization function which aims to avoid out-of-distribution actions. The third constraint ensures that the extracted policy is the policy implied in $Q, V$.

Here we briefly describe the characteristics of the solution to problem IPF. In problem IPF, when the feasible set includes multiple policies (*i.e.* multiple implicit policies satisfy Definition 4.1), problem IPF aims to find an optimal implicit policy that deviates least from the behavior policy while satisfying the requirements of policy alignment. In other cases, when the feasible set has a unique policy, problem IPF will return the unique policy as the optimal implicit policy. The above analysis shows that we can model the implicit policy-finding problem in IQL as problem IPF.

**Assumption 4.2.** Assume $\pi(\boldsymbol{a}|\boldsymbol{s}) > 0 \implies \mu(\boldsymbol{a}|\boldsymbol{s}) > 0$ so that $\frac{\pi(\boldsymbol{a}|\boldsymbol{s})}{\mu(\boldsymbol{a}|\boldsymbol{s})}$ is well-defined. (Xu et al., 2023)

**Assumption 4.3.** Assume that $f(x)$ is differentiable on $(0, \infty)$ and that $h_f(x) = xf(x)$ is strictly convex and $f(1) = 0$. (Xu et al., 2023)

*Remark* 4.4. Under the above assumptions, problem IPF is a convex optimization problem and assumption 4.3 makes the regularization term positive due to Jensen's inequality as $\mathbb{E}_\mu[\frac{\pi}{\mu}f(\frac{\pi}{\mu})] \geq 1, f(1) = 0$ (Xu et al., 2023). Slater's conditions hold since the first and second constraints define a probability simplex, and the third constraint defines a hyperplane in the tabular setting. The intersection of these convex sets is nonempty if the optimal policy exists, *i.e.* the optimal policy is not a uniform distribution. The analysis described above shows that this convex optimization problem is feasible and Slater's conditions are satisfied.

## 5 OPTIMIZATION

In this section, we introduce two methods AlignIQL-hard and AlignIQL for solving problem IPF. Theoretically, AlignIQL-hard is more rigorous as it strictly ensures policy alignment and provides insights into why IQL can use AWR for policy extraction, but it suffers from complex training. AlignIQL avoids the training complexity of AlignIQL-hard while also guaranteeing local convergence to the optimal solution of problem IPF through soft constraints. All proof can be found in Appendix A.

### 5.1 ALIGNIQL-HARD

We first consider directly solving IPF with KKT conditions (See proof in Appendix A.1) and get the following theorems.

**Theorem 5.1.** *For problem IPF, the optimal policy $\pi^*$ and its optimal Lagrange multipliers satisfy the following optimality condition for all states and actions:*

$$\pi^\star(\boldsymbol{a}|\boldsymbol{s}) = \mu(\boldsymbol{a}|\boldsymbol{s}) \max \{g_f(-\alpha^*(\boldsymbol{s}) - \beta^*(\boldsymbol{s})Q(\boldsymbol{s},\boldsymbol{a})), 0\}. \tag{9}$$

$$\mathbb{E}_{\boldsymbol{a} \sim \mu} [\max \{g_f(-\alpha^*(\boldsymbol{s}) - \beta^*(\boldsymbol{s})Q(\boldsymbol{s},\boldsymbol{a})), 0\}] = 1, \tag{10}$$

$$\mathbb{E}_{\boldsymbol{a} \sim \mu(\boldsymbol{a}|\boldsymbol{s})} [Q(\boldsymbol{s},\boldsymbol{a}) \max \{g_f(-\alpha^*(\boldsymbol{s}) - \beta^*(\boldsymbol{s})Q(\boldsymbol{s},\boldsymbol{a})), 0\} - V(\boldsymbol{s})] = 0, \tag{11}$$

*where $\alpha^*, \beta^*$ is the Lagrange multiplier, $g_f$ is the inverse function of $h'_f(x)$.*

We can also follow Peters et al. (2010); Peng et al. (2019); Nair et al. (2020) to train our policy $\pi_\phi$ through

$$\arg\min_\phi \mathbb{E}_{\boldsymbol{s} \sim \mathcal{D}^\mu} [D_{\mathrm{KL}} (\pi^*(\cdot|\boldsymbol{s})||\pi_\phi(\cdot|\boldsymbol{s}))]$$

$$\approx \arg\min_\phi \mathcal{L}_\pi(\phi) = \mathbb{E}_{(\boldsymbol{s},\boldsymbol{a}) \sim \mathcal{D}} [-\max \{g_f(-\alpha^*(\boldsymbol{s}) - \beta^*(\boldsymbol{s})Q(\boldsymbol{s},\boldsymbol{a})), 0\} \log \pi_\phi(\boldsymbol{a}|\boldsymbol{s})]. \tag{12}$$

However, loss function Equation 12 needs the exact policy density, which may limit the usage of diffusion models or other generative models.

*Remark* 5.2. Note that $\alpha^*$ is a normalization term, it does not affect the action generated by the policy. Let $f(x) = \log x$, then $g_f(x) = \exp(x - 1)) > 0$, we can get $\pi^*(\boldsymbol{a}|\boldsymbol{s}) \propto \mu(\boldsymbol{a}|\boldsymbol{s}) \exp(-\beta^*Q(\boldsymbol{s},\boldsymbol{a}))$ In most environments (especially MuJoCo tasks), $\beta^*$ we learned through the neural network is negative. Because only the positive and negative of $\beta^*$ affect the action generated by the policy, we can approximate $-\beta^*$ with a fixed $\beta \in (0, \infty]$, *i.e.* $\pi^*(\boldsymbol{a}|\boldsymbol{s}) \propto \mu(\boldsymbol{a}|\boldsymbol{s}) \exp(\beta Q(\boldsymbol{s},\boldsymbol{a}))$, which is exactly what optimal policy obtained by AWR. This explains why IQL can learn implicit policy with weighted regression and shows implicit policy further avoids the OOD actions through the regularization function $f$, which gives a deeper understanding of how IQL-style methods handle the distribution shift. This also addresses the issue in IDQL, that is, they find that simply taking the action with the highest Q-value usually yields better performance at evaluation time.

Previous works (Hansen-Estruch et al., 2023; Chen et al., 2022) often use the increasing function of $Q(\boldsymbol{s},\boldsymbol{a})$ as a weight. However, according to Theorem 5.1, when $\beta^*(\boldsymbol{s}) \geq 0$, we need to be more conservative, that is, we should choose actions with lower $Q(\boldsymbol{s},\boldsymbol{a})$. To calculate the weights, we need to solve the closed-form solution of Equation 10, Equation 11, which is usually intractable. However, we can use the parameterized neural network to approximate it.

**Lemma 5.3.** *Following EQL (Xu et al., 2023), let $f(x) = \log x$, then $g_f(x) = \exp(x - 1)) > 0$. We can approximate $\alpha^*(s)$, $\beta^*(s)$ through neural network with the following loss function:*

$$\max_{\alpha,\beta} \mathcal{L}_M = -\mathbb{E}_{\boldsymbol{a}\sim\mu}\left[\exp\left(-\alpha(\boldsymbol{s}) - \beta(\boldsymbol{s})Q(\boldsymbol{s},\boldsymbol{a}) - 1\right)\right] - \alpha(\boldsymbol{s}) - \beta(\boldsymbol{s})V(\boldsymbol{s}), \tag{13}$$

*Proof.* Then Lemma 5.3 can be get through setting the gradient of Equation 13 to 0 with respect to $\alpha, \beta$, which is Equation 10, Equation 11 respectively. $\qquad\square$

*Remark* 5.4. Now we can obtain $\alpha^*, \beta^*$ by iteratively updating $\alpha, \beta$ following Equation 13.

Based on Theorem 5.1 and Lemma 5.3, we can get AlignIQL-hard, where hard means we rigidly constrain the policy to satisfy policy alignment. AlignIQL-hard shows when multiplier $\beta(\boldsymbol{s}) < 0$, we can use AWR for extracting the implicit policy in IQL. However, for strict policy alignment, AlignIQL-hard needs to train an additional two multiplier networks, which increases the training costs and compound errors. Moreover, the exponential term in Equation 13 makes the unstable training. In the remainder of this section, we introduce a more simple and effective method AlignIQL to solve problem IPF.

## 5.2 ALIGNIQL

In this section, we introduce AlignIQL to solve the alignment problem of IQL. Firstly, we introduce the soft constraint form of problem IPF. Given $\eta > 0$ , IPF-Soft is defined as

$$\min_{\pi,V(\boldsymbol{s})} \quad \mathbb{E}_{\boldsymbol{s}\sim d^\pi(\boldsymbol{s}),\boldsymbol{a}\sim\pi(\boldsymbol{a}|\boldsymbol{s})}\left[f\left(\frac{\pi(\boldsymbol{a}|\boldsymbol{s})}{\mu(\boldsymbol{a}|\boldsymbol{s})}\right) + \eta\left(Q(\boldsymbol{s},\boldsymbol{a}) - V(\boldsymbol{s})\right)^2\right]$$

$$s.t. \quad \pi(\boldsymbol{a}|\boldsymbol{s}) \geq 0, \quad \forall \boldsymbol{s}, \forall \boldsymbol{a} \tag{IPF-Soft}$$

$$\int_{\boldsymbol{a}} \pi(\boldsymbol{a}|\boldsymbol{s})d\boldsymbol{a} = 1, \quad \forall \boldsymbol{s}.$$

*Remark* 5.5. Note that we relax problem IPF by adding penalty term $\mathbb{E}_{\boldsymbol{a}\sim\pi(\boldsymbol{a}|\boldsymbol{s})}[\eta\left(Q(\boldsymbol{s},\boldsymbol{a}) - V(\boldsymbol{s})\right)^2]$ rather than $\eta(\mathbb{E}_{\boldsymbol{a}\sim\pi(\boldsymbol{a}|\boldsymbol{s})}[Q(\boldsymbol{s},\boldsymbol{a})] - V(\boldsymbol{s}))^2$. The latter relaxation formulation is equivalent to the quadratic penalty method, whose convergence relies on the penalty parameter $\eta$ approaching positive infinity which leads to an ill-conditioned Hessian matrix for the quadratic penalty function (Nocedal & Wright, 1999). Our penalty term can avoid this issue since the optimal solution of $\mathbb{E}_{\boldsymbol{a}\sim\pi(\boldsymbol{a}|\boldsymbol{s})}[\eta\left(Q(\boldsymbol{s},\boldsymbol{a}) - V(\boldsymbol{s})\right)^2]$ satisfies Equation 8 (setting the gradient to 0 with respect to $V$), which shows that our penalty term can implicitly recover policy alignment constraint Equation 8.

We refer to the above problem as problem IPF-Soft, since the policy alignment is not rigidly held. Then we solve problem IPF-Soft by KKT conditions (See proof in Appendix A.2) and get the optimal policy:

$$\pi^\star(\boldsymbol{a}|\boldsymbol{s}) = \mu(\boldsymbol{a}|\boldsymbol{s})\max\left\{g_f\left(-\alpha(\boldsymbol{s}) - \eta\left(Q(\boldsymbol{s},\boldsymbol{a}) - V(\boldsymbol{s})\right)^2\right), 0\right\}. \tag{14}$$

**Theorem 5.6.** *Suppose that $f(x) = \log x$, then the optimal policy of problem IPF-Soft satisfies*

$$\pi^\star(\boldsymbol{a}|\boldsymbol{s}) \propto \mu(\boldsymbol{a}|\boldsymbol{s})\exp\left\{-\eta\left(Q(\boldsymbol{s},\boldsymbol{a}) - V(\boldsymbol{s})\right)^2\right\}. \tag{15}$$

*If the exact policy density is known, we can also follow Peters et al. (2010); Peng et al. (2019); Nair et al. (2020) to train our policy $\pi_\phi$ through*

$$\arg\min_{\phi} \mathbb{E}_{\boldsymbol{s}\sim\mathcal{D}^\mu}\left[D_{\text{KL}}\left(\pi^*(\cdot|\boldsymbol{s})||\pi_\phi(\cdot|\boldsymbol{s})\right)\right]$$

$$\approx \arg\min_{\phi} \mathcal{L}_\pi(\phi) = \mathbb{E}_{(\boldsymbol{s},\boldsymbol{a})\sim\mathcal{D}}\left[-\exp\left(-\eta\left(Q(\boldsymbol{s},\boldsymbol{a}) - V(\boldsymbol{s})\right)^2\right)\log\pi_\phi(\boldsymbol{a}|\boldsymbol{s})\right]. \tag{16}$$

**Compared to AWR:** For $\eta > 0$, Equation 15 prefers actions that minimize $(Q(\boldsymbol{s},\boldsymbol{a}) - V(\boldsymbol{s}))^2$. This is different from AWR, which prefers actions with higher $Q(\boldsymbol{s},\boldsymbol{a})$. The reason behind this difference is that AlignIQL aims to balance between behavior cloning and policy alignment, whereas AWR

aims to balance between behavior cloning and critic exploitation. In fact, $\eta$ can also be interpreted as implicit critic exploitation. In IQL, $V(\boldsymbol{s})$ is trained to approximate $\arg\max_{\boldsymbol{a}\sim\mathcal{D}} Q(\boldsymbol{s},\boldsymbol{a})$. A higher $\eta$ increases the probability assigned to actions where $Q(\boldsymbol{s},\boldsymbol{a}) \approx V(\boldsymbol{s})$, which can be viewed as an implicit policy improvement, as $V(\boldsymbol{s}) = \arg\max_{\boldsymbol{a}\sim\mathcal{D}} Q(\boldsymbol{s},\boldsymbol{a})$ in IQL.

Finally, we show the connection between the solution of problem IPF and problem IPF-Soft through the following Proposition 5.7.

**Proposition 5.7.** *Suppose that $\pi^*(\boldsymbol{a}|\boldsymbol{s})$ is a global solution to the convex optimization problem IPF, with its corresponding value function (denoted as $V^*(\boldsymbol{s})$). Then there exists a $\eta$ such that $\pi^*, V^*(\boldsymbol{s})$ is a local minimizer of problem IPF-Soft. (See proof in Appendix A.3.)*

*Remark* 5.8. Proposition 5.7 indicates that we can obtain the solution to problem IPF by solving problem Equation IPF-Soft. Because KKT conditions are the first-order necessary for a solution in nonlinear programming to be optimal, the solution to problem IPF can be written in the form of Equation 15. This implies that if we train $Q(\boldsymbol{s},\boldsymbol{a})$ and $V(\boldsymbol{s})$ using IQL and $\eta$ satisfies Proposition 5.7, we can extract the implicit policy from the value function using Equation 15. Actually, for IQL, the expectile loss Equation 4 approximates the maximum of $Q_{\hat{\theta}}(\boldsymbol{s},\boldsymbol{a})$ when $\tau \approx 1$. We can approximately think $V(\boldsymbol{s}) = \arg\max_{\boldsymbol{a}\sim\mathcal{D}} Q(\boldsymbol{s},\boldsymbol{a})$, and thus, according to Equation 15, $\hat{\boldsymbol{a}} = \arg\max_{\boldsymbol{a}} Q(\boldsymbol{s},\boldsymbol{a})$ has a weight of 1, while other actions are weighted by $\exp\left\{-\eta(Q(\boldsymbol{s},\boldsymbol{a}) - V(\boldsymbol{s}))^2\right\}$. For a fixed $\eta$, the weights for other actions are smaller than $\arg\max_{\boldsymbol{a}\sim\mathcal{D}} Q(\boldsymbol{s},\boldsymbol{a})$. Therefore, Equation 15 approximately recovers the implicit policy $\pi^*(\boldsymbol{a}|\boldsymbol{s}) = \arg\max_{\boldsymbol{a}\sim\mathcal{D}} Q^*(\boldsymbol{s},\boldsymbol{a})$ from IQL learned value functions.

**Two ways to use AlignIQL or AlignIQL-hard:** There are two ways to utilize our methods in offline RL (corresponding to Algorithm 3 and Algorithm 1).

- **Energy-based implementation:** We first use the learned diffusion-based behavior model $\mu_\phi(\boldsymbol{a}|\boldsymbol{s})$ to generate $N$ action samples. These actions are then evaluated using weights from Equation 15 or Equation 9 (Algorithm 3). In this setting, the hyperparameter $N$ has a greater influence on performance than $\eta$, as a higher $N$ is more likely to find the "lucky" action that satisfies $\hat{\boldsymbol{a}} = \arg\max_{\boldsymbol{a}} Q(\boldsymbol{s},\boldsymbol{a})$.

- **Policy-based implementation:** We use Equation 16 or Equation 12 to train the policy, which needs the exact probability density of the current policy (Algorithm 1).

In summary, the first method can be used when employing diffusion-based policies, as the probability density of diffusion models is unknown. The second method is applicable when using Gaussian-based policies. Note that in both AlignIQL-hard and AlignIQL, we do not impose a limit on the loss function of the $Q - V$, which means that our conclusion can be generalized to the arbitrary critic loss function and the arbitrary sub-optimal value function. To summarize, both the AlignIQL-hard and AlignIQL are "IQL-style" algorithms, which means the training of actor and critic are decoupled and the critic is learned by expectile regression. The difference between AlignIQL-hard and AlignIQL lies in the calculation of weights and the necessity of training multiplier networks. We summarize the procedure of AlignIQL-hard and AlignIQL in Algorithm 2 and Algorithm 3. (Suppose that $f(x) = \log x$.)

# 6 EXPERIMENTS

In this section, we conduct extensive experiments and specifically answer the following questions

- **Q1**: Can AlignIQL match the performance with other SOTA offline RL baselines?
- **Q2**: What are the benefits of using weights from AlignIQL?
- **Q3**: Both IDQL and AlignIQL derive the weights needed for policy extraction. Are our weights better than IDQL?

## 6.1 RESULTS ON D4RL TASKS (Q1)

We conduct extensive experiments on D4RL datasets (Fu et al., 2020) to verify the performance of AlignIQL.

**Baselines:** For the selection of baselines, we select Conservative Q-learning (CQL) (Kumar et al., 2020), DiffusionQL (Wang et al., 2022), Implicit-Q Learning (IQL) (Kostrikov et al., 2021a), SQL (Xu et al., 2023), SfBC (Chen et al., 2022), which first trains a diffusion-based policy and then selects actions based on the $Q$ value, similar to AWR, because of their strong performance in the offline RL setting. We also select the DD (Ajay et al., 2022) and Diffuser (Janner et al., 2022) as baselines since they represent sequential-modeling-based RL algorithms.

Aggregated results can be found in Table 1 and Table 2. In MuJoCo tasks, where the performance is already saturated, therefore we use the policy-based implementation of AlignIQL (abbreviated AlignIQL) for faster training. Additionally, we noticed that energy-based implementation shows slightly worse results than other methods.(Table 4) This may be due to the saturated performance in MuJoCo tasks, where the impact of policy alignment is less pronounced, and the objective function in Equation IPF restricts exploration.

In more challenging AntMaze tasks, we use the energy-based implementation of AlignIQL (abbreviated D-AlignIQL). D-AlignIQL outperforms other methods by a large margin in Antmaze tasks Table 2. More importantly, we find that D-AlignIQL benefits from larger values of $N$, whereas larger $N$ often leads to performance degradation in other diffusion-based methods such as IDQL. We will elaborate on this in the following sections.

Table 1: The performance of our method and other SOTA baselines on MuJoCo tasks. We use the policy-based implementation of AlignIQL. The best result is highlighted in SkyBlue. We report the score of AlignIQL by choosing the best scores from $\eta \in \{0.5, 1, 5, 10\}$ over 3 random seeds.

| Dataset | Environment | CQL | Diffusion-QL | SfBC | SQL | DD | Diffuser | IDQL | IQL | AlignIQL (ours) |
|---|---|---|---|---|---|---|---|---|---|---|
| Medium-Expert | HalfCheetah | 62.4 | 96.8 | 92.6 | 94.0 | 90.6 | 79.8 | 89.2 | 86.7 | 84.6±0.36 |
| Medium-Expert | Hopper | 98.7 | 111.1 | 108.6 | 111.8 | 111.8 | 107.2 | 108.2 | 91.5 | 97.7±3.9 |
| Medium-Expert | Walker2d | 111.0 | 110.1 | 109.8 | 110.0 | 108.8 | 108.4 | 111.7 | 109.6 | 110.5±0.03 |
| Medium | HalfCheetah | 44.4 | 51.1 | 45.9 | 48.3 | 49.1 | 44.2 | 46.0 | 47.4 | 42.7±0.02 |
| Medium | Hopper | 58.0 | 90.5 | 57.1 | 75.5 | 79.3 | 58.5 | 56.3 | 66.3 | 69.1±0.4 |
| Medium | Walker2d | 79.2 | 87.0 | 77.9 | 84.2 | 82.5 | 79.7 | 77.6 | 78.3 | 83.2±0.1 |
| Medium-Replay | HalfCheetah | 46.2 | 47.8 | 37.1 | 44.8 | 39.3 | 42.2 | 41.1 | 44.2 | 45.1±0.01 |
| Medium-Replay | Hopper | 48.6 | 101.3 | 86.2 | 99.7 | 100.0 | 96.8 | 86.2 | 94.7 | 91.1±4.4 |
| Medium-Replay | Walker2d | 26.7 | 95.5 | 65.1 | 81.2 | 75.0 | 61.2 | 85.1 | 73.9 | 82.2±8.8 |
| **Average (Locomotion)** | | 63.9 | 87.9 | 75.6 | 83.3 | 81.8 | 75.3 | 78.0 | 76.9 | 78.5 |
| **# Diffusion steps** | | - | 5 | 15 | — | 100 | 100 | 5 | - | — |

Table 2: The performance of our method and other SOTA baselines on AntMaze tasks. We report the performance of baseline methods using the best results reported from their papers except for IDQL. We rerun the official code of IDQL and report the results on the same hardware (RTX 4090 with 24GB memory) for a fair comparison. We report the best result of D-AlignIQL and IDQL by taking the average of the last evaluation over 10 seeds. The best result is highlighted in SkyBlue. "D-" means diffusion-based implementation.

| Dataset | Environment | CQL | Diffusion-QL | SfBC | SQL | DD | Diffuser | IDQL | IQL | D-AlignIQL (ours) |
|---|---|---|---|---|---|---|---|---|---|---|
| Default | AntMaze-umaze | 74.0 | 93.4 | 92.0 | 92.2 | - | - | 93.4 | 87.5 | 94.8±3.2 |
| Diverse | AntMaze-umaze | 84.0 | 66.2 | 85.3 | 74.0 | - | - | 75.2 | 62.2 | 82.4±4.4 |
| Play | AntMaze-medium | 61.2 | 76.6 | 81.3 | 80.2 | - | - | 85 | 71.2 | 87.5±2.5 |
| Diverse | AntMaze-medium | 53.7 | 78.6 | 82.0 | 79.1 | - | - | 74.4 | 70.0 | 85.0±5.0 |
| Play | AntMaze-large | 15.8 | 46.4 | 59.3 | 53.2 | - | - | 60.0 | 39.6 | 65.2±9.6 |
| Diverse | AntMaze-large | 14.9 | 57.3 | 45.5 | 52.3 | - | - | 58.4 | 47.5 | 66.4±9.7 |
| **Average (AntMaze)** | | 50.6 | 69.8 | 74.2 | 71.8 | - | - | 74.4 | 63.0 | 80.2 |
| **# Diffusion steps** | | - | 5 | 15 | — | 100 | 100 | 5 | - | 5 |

## 6.2 COMPARISON WITH DIFFUSION+AWR (Q2)

In this section, we conduct a detailed comparison of the impact of the policy alignment. We first train a diffusion-based behavior policy, then we sample $N$ actions from the diffusion-based behavior policy and select the action with maximum $Q(s, a)$, where the value function is learned by IQL.

We choose the AntMaze tasks since challenging tasks can better show the effect of alignment. In the experiment shown in Figure 1, we use a diffusion-based behavior policy and guarantee that all the hyperparameters and network structures are identical except for parameters used for policy extraction. Figure 1 shows that under the same $N$ ( where $N$ represents the samples per state), The correctly aligned policy (D-AlignIQL in Figure 1) converges faster and does not experience a significant performance drop at higher $N = 256$. Generally, the greedy selection of actions based on Q-values (Brandfonbrener et al., 2021; Haarnoja et al., 2018b), which usually yields better results, but performs poorly here. This is due to AWR-based weights assigning high values to out-of-distribution actions potentially generated by the behavior policy. In MuJoCo tasks, due to the relative simplicity of the tasks, the Q-function can achieve good generalization performance through training, so this issue is not pronounced. Above all, the results in Figure 1 show the benefits of policy alignment and demonstrate the correctness of our theory through the performance difference between D-AlignIQL and AWR .

### 6.3 COMPARISON WITH IDQL (Q2,Q3)

Since both IDQL and D-AlignIQL provide weights under policy alignment, in this part we evaluate which weights are better through D4RL AntMaze tasks and ablation study on $N$. In this section's experiments, IDQL and D-AlignIQL use expectile regression to learn the critic network. The details of weight used by IDQL can be found in Appendix D. We use expectile regression to train the critic network since the expectile objective is used in IQL. Of course, our D-AlignIQL framework can also be extended to other generalized critic loss functions mentioned in IDQL.

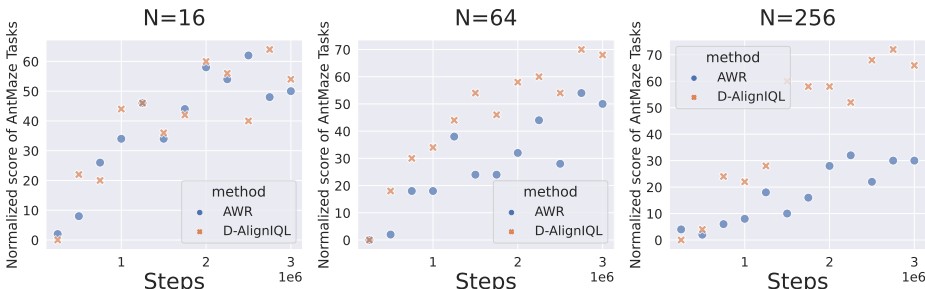

Figure 1: Performance of D-AlignIQL and AWR (Diffusion-based implementation) on AntMaze tasks across different training steps. The horizontal axis represents time (s) on a logarithmic scale. Results are averaged over 10 random seeds.

**Ablation Study**. Since the main limitation of diffusion-based methods is the running speed, we compare the performance of IDQL and D-AlignIQL at different training times in Figure 2. Figure 2 shows that as training time increases, the performance of AlignIQL improves with increasing $N$, whereas IDQL does not exhibit a similar trend. This phenomenon demonstrates the robustness of our method, as we expect that, for a robust method, the performance with different values of $N$ should not degrade. We also observe that D-AlignIQL converges faster than IDQL. In Appendix F.2, we also compare the D-AlignIQL with other methods in terms of running time and show that the training time of D-AlignIQL can be significantly reduced through acceleration methods (Kang et al., 2024; Lu et al., 2022). In Appendix H, we also compare the effect of different regularizers function $f(x)$. Overall, compared to IDQL, the weights computed by our method not only have better theoretical properties (applicable to any Q-loss, without requiring optimal $V$) but also perform better in practice. We also compare D-AlignIQL-hard (Diffusion-based AlignIQL-hard) and D-AlignIQL in Appendix F.1.

We also present the quantitative scores of D-AlignIQL and IDQL on AntMaze tasks (Figure 2) to highlight the superiority of D-AlignIQL.

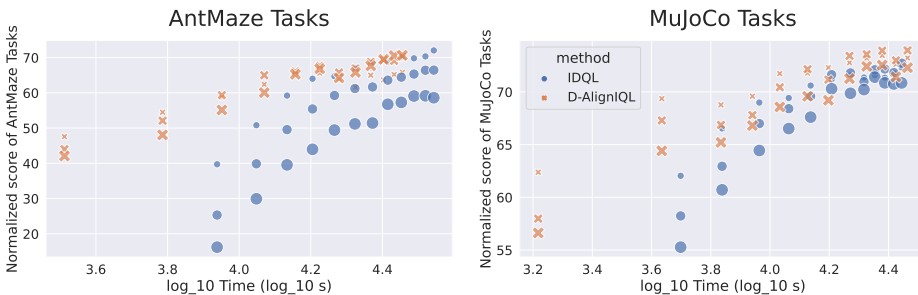

Figure 2: Performance of D-AlignIQL and IDQL on AntMaze tasks with different training steps. The different sizes, from smallest to largest, represent $N = 16, N = 64, N = 256$, respectively.

Table 3: Quantitative Results of D-AlignIQL and IDQL on AntMaze Large tasks.

| | IDQL | | | D-AlignIQL | | |
|---|---|---|---|---|---|---|
| D4RL Tasks | $N = 16$ | $N = 64$ | $N = 256$ | $N = 16$ | $N = 64$ | $N = 256$ |
| **AntMaze** | 72.0 | 66.5 | 58.8 | 65.8 | 70.2 | 70.7 |

As shown in Table 3, the performance of D-AlignIQL improves with increasing $N$, whereas IDQL does not exhibit such a trend. This is because our method selects actions to minimize $(Q - V)^2$, and when $V$ approaches the optimal value $V^*(\boldsymbol{s}) = \max_{\boldsymbol{a} \sim \mathcal{D}} Q(\boldsymbol{s}, \boldsymbol{a})$, D-AlignIQL equals to select $\boldsymbol{a} = \arg\max_{\boldsymbol{a} \sim \mathcal{D}} Q(\boldsymbol{s}, \boldsymbol{a})$. Compared to other weights, such as AWR, which directly selects actions based on $Q$-values, D-AlignIQL is more robust to variations in $N$. This robustness arises because out-of-distribution (OOD) actions generated by the policy network (e.g., using a diffusion model) are unlikely to exactly match $V(\boldsymbol{s})$ but may still exhibit higher $Q(\boldsymbol{s}, \boldsymbol{a})$. (Fujimoto et al., 2018). This explains why our method benefits from larger $N$.

## 7 DISCUSSION

**Discussion.** In our work, we define the implicit policy-finding problem in IQL and propose two practical algorithms AlignIQL-hard and AlignIQL to solve it. The optimal policy (Theorem 5.1) in AlignIQL-hard shows that it is feasible to extract policy with AWR in certain cases, which builds the bridge between the Implicit Q-learning and Weighted Regression. Our theoretical findings also extend the policy alignment of IDQL to arbitrary critic loss and value functions. Besides the theoretical findings, we also verify the effectiveness of our algorithm on D4RL datasets. Experimental results show that compared to other IQL-style algorithms, our algorithm achieves SOTA performance and is more stable, especially in sparse reward tasks. One future work is to explore better methods for training multiplier networks and explore the impact of different regularization functions of problem IPF. Another future work is to extend our approach to fields of safe RL and offline-to-online (O2O) learning. In safe RL, prior works (Zheng et al., 2023; Cao et al., 2024) have used IQL to learn the Q-function. Investigating how to ensure policy alignment while satisfying safety constraints is an interesting research direction.

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

# A    PROOF

## A.1    PROOF OF THEOREM 5.1

*Proof.* The Lagrange function of Equation IPF is written as follows

$$L(\pi, \alpha(\boldsymbol{s}), \beta(\boldsymbol{s}), \lambda) = \mathbb{E}_{\boldsymbol{s}\sim d^{\pi}(\boldsymbol{s}), \boldsymbol{a}\sim\pi(\boldsymbol{a}|\boldsymbol{s})}\left[f(\frac{\pi(\boldsymbol{a}|\boldsymbol{s})}{\mu(\boldsymbol{a}|\boldsymbol{s})})\right] - \mathbb{E}_{\boldsymbol{s}\sim d^{\pi}(\boldsymbol{s}), \boldsymbol{a}\sim\pi(\boldsymbol{a}|\boldsymbol{s})}\left[\lambda(\boldsymbol{a}|\boldsymbol{s})\pi(\boldsymbol{a}|\boldsymbol{s})\right]$$

$$+ \mathbb{E}_{\boldsymbol{s}\sim d^{\pi}(\boldsymbol{s})}\left[\alpha(\boldsymbol{s})\left(\int_{\boldsymbol{a}}\pi(\boldsymbol{a}|\boldsymbol{s})d\boldsymbol{a} - 1\right)\right] + \tag{17}$$

$$\mathbb{E}_{\boldsymbol{s}\sim d^{\pi}(\boldsymbol{s})}\left[\beta(\boldsymbol{s})\left(\mathbb{E}_{\boldsymbol{a}\sim\pi(\boldsymbol{a}|\boldsymbol{s})}\left[Q(\boldsymbol{s}, \boldsymbol{a})\right] - V(\boldsymbol{s})\right)\right],$$

where $d^{\pi}(\boldsymbol{s})$ represents the state distribution induced by policy $\pi$, $\alpha(\boldsymbol{s})$, $\beta(\boldsymbol{s})$, and $\lambda$ are Lagrangian multipliers for the equality and inequality constraints respectively.

Let $h_f(x) = xf(x)$. Then for all states and actions, the KKT conditions can be written as follows

$$\pi(\boldsymbol{a}|\boldsymbol{s}) \geq 0 \tag{18}$$

$$\int_{\boldsymbol{a}}\pi(\boldsymbol{a}|\boldsymbol{s})d\boldsymbol{a} = 1 \tag{19}$$

$$\mathbb{E}_{\boldsymbol{a}\sim\pi(\boldsymbol{a}|\boldsymbol{s})}\left[Q(\boldsymbol{s}, \boldsymbol{a}) - V(\boldsymbol{s})\right] = 0 \tag{20}$$

$$\lambda(\boldsymbol{a}|\boldsymbol{s}) \geq 0 \tag{21}$$

$$\lambda(\boldsymbol{a}|\boldsymbol{s})\pi(\boldsymbol{a}|\boldsymbol{s}) = 0 \tag{22}$$

$$h'_f(\frac{\pi(\boldsymbol{a}|\boldsymbol{s})}{\mu(\boldsymbol{a}|\boldsymbol{s})}) + \alpha(\boldsymbol{s}) + \beta(\boldsymbol{s})Q(\boldsymbol{s}, \boldsymbol{a}) - \lambda(\boldsymbol{a}|\boldsymbol{s}) = 0 \tag{23}$$

We eliminate $d^{\pi}(\boldsymbol{s})$ due to irreducible Markov chain assumption. Note that in our derivation, we assume that $V(\boldsymbol{s})$ and $Q(\boldsymbol{s}, \boldsymbol{a})$ are known.

Since $h'_f$ is a strictly increasing function, its inverse function exists and is also a strictly increasing function. Let $g_f = (h'_f)^{-1}(x)$ be its inverse function. From Equation 23, we can get

$$\pi(\boldsymbol{a}|\boldsymbol{s}) = \mu(\boldsymbol{a}|\boldsymbol{s})g_f\left(\lambda(\boldsymbol{a}|\boldsymbol{s}) - \alpha(\boldsymbol{s}) - \beta(\boldsymbol{s})Q(\boldsymbol{s}, \boldsymbol{a})\right) \tag{24}$$

Given a state $\boldsymbol{s}$, we can get $\lambda(\boldsymbol{a}|\boldsymbol{s}) = h'_f(\frac{\pi}{\mu}) + \alpha(\boldsymbol{s}) + \beta(\boldsymbol{s})Q(\boldsymbol{s}, \boldsymbol{a})$ from Equation 23, then

(a) If $\lambda(\boldsymbol{a}|\boldsymbol{s}) = h'_f(\frac{\pi}{\mu}) + \alpha(\boldsymbol{s}) + \beta(\boldsymbol{s})Q(\boldsymbol{s}, \boldsymbol{a}) > 0$, then $\pi(\boldsymbol{a}|\boldsymbol{s})$ is zero due to complementary slackness. Note that $\pi(\boldsymbol{a}|\boldsymbol{s}) = 0$, thus $h'_f(0) + \alpha(\boldsymbol{s}) + \beta(\boldsymbol{s})Q(\boldsymbol{s}, \boldsymbol{a}) > 0$ and we can get $g_f(-\alpha(\boldsymbol{s}) - \beta(\boldsymbol{s})Q(\boldsymbol{s}, \boldsymbol{a})) < g_f(h'_f(0)) = 0$.

(b) If $\lambda(\boldsymbol{a}|\boldsymbol{s}) = 0$, then $h'_f(\frac{\pi}{\mu}) + \alpha(\boldsymbol{s}) + \beta(\boldsymbol{s})Q(\boldsymbol{s}, \boldsymbol{a})$ is zero and $\pi(\boldsymbol{a}|\boldsymbol{s}) = \mu(\boldsymbol{a}|\boldsymbol{s})g_f\left(-\alpha(\boldsymbol{s}) - \beta(\boldsymbol{s})Q(\boldsymbol{s}, \boldsymbol{a})\right) \geq 0$. Note that $\pi(\boldsymbol{a}|\boldsymbol{s}) \geq 0$, thus $h'_f(0) + \alpha(\boldsymbol{s}) + \beta(\boldsymbol{s})Q(\boldsymbol{s}, \boldsymbol{a}) \leq 0$ and we can get $g_f(-\alpha(\boldsymbol{s}) - \beta(\boldsymbol{s})Q(\boldsymbol{s}, \boldsymbol{a})) \geq g_f(h'_f(0)) = 0$.

Through analysis (a) and (b), we can resolve optimal policy $\pi^*(\boldsymbol{a}|\boldsymbol{s})$ as

$$\pi^\star(\boldsymbol{a}|\boldsymbol{s}) = \mu(\boldsymbol{a}|\boldsymbol{s})\max\left\{g_f\left(-\alpha(\boldsymbol{s}) - \beta(\boldsymbol{s})Q(\boldsymbol{s}, \boldsymbol{a})\right), 0\right\}. \tag{25}$$

Substituting back in Equation 19 and Equation 20 with Equation 9, we can get

$$\mathbb{E}_{\boldsymbol{a}\sim\mu}\left[\max\left\{g_f(-\alpha^*(\boldsymbol{s}) - \beta^*(\boldsymbol{s})Q(\boldsymbol{s}, \boldsymbol{a})), 0\right\}\right] = 1, \tag{26}$$

$$\mathbb{E}_{\boldsymbol{a}\sim\mu(\boldsymbol{a}|\boldsymbol{s})}\left[Q(\boldsymbol{s}, \boldsymbol{a})\max\left\{g_f(-\alpha^*(\boldsymbol{s}) - \beta^*(\boldsymbol{s})Q(\boldsymbol{s}, \boldsymbol{a})), 0\right\} - V(\boldsymbol{s})\right] = 0, \tag{27}$$

$\square$

### A.2 PROOF OF THEOREM 5.6

*Proof.* The Lagrange function of Equation IPF-Soft is written as follows

$$
\begin{aligned}
L(\pi, V, \alpha(\boldsymbol{s}), \beta(\boldsymbol{s}), \lambda) = {} & \mathbb{E}_{\boldsymbol{s} \sim d^\pi(\boldsymbol{s}), \boldsymbol{a} \sim \pi(\boldsymbol{a}|\boldsymbol{s})} \left[ f(\frac{\pi(\boldsymbol{a}|\boldsymbol{s})}{\mu(\boldsymbol{a}|\boldsymbol{s})}) + \eta \left(Q(\boldsymbol{s}, \boldsymbol{a}) - V(\boldsymbol{s})\right)^2 \right] \\
& - \mathbb{E}_{\boldsymbol{s} \sim d^\pi(\boldsymbol{s}), \boldsymbol{a} \sim \pi(\boldsymbol{a}|\boldsymbol{s})} \left[\lambda(\boldsymbol{a}|\boldsymbol{s})\pi(\boldsymbol{a}|\boldsymbol{s})\right] \\
& + \mathbb{E}_{\boldsymbol{s} \sim d^\pi(\boldsymbol{s})} \left[\alpha(\boldsymbol{s}) \left(\int_{\boldsymbol{a}} \pi(\boldsymbol{a}|\boldsymbol{s})d\boldsymbol{a} - 1\right)\right].
\end{aligned}
\tag{28}
$$

Let $h_f(x) = xf(x)$. Then for all states and actions, the KKT conditions can be written as follows

$$
\pi(\boldsymbol{a}|\boldsymbol{s}) \geq 0
\tag{29}
$$

$$
\int_{\boldsymbol{a}} \pi(\boldsymbol{a}|\boldsymbol{s})d\boldsymbol{a} = 1
\tag{30}
$$

$$
\mathbb{E}_{\boldsymbol{a} \sim \pi(\boldsymbol{a}|\boldsymbol{s})} \left[Q(\boldsymbol{s}, \boldsymbol{a}) - V(\boldsymbol{s})\right] = 0
\tag{31}
$$

$$
\lambda(\boldsymbol{a}|\boldsymbol{s}) \geq 0
\tag{32}
$$

$$
\lambda(\boldsymbol{a}|\boldsymbol{s})\pi(\boldsymbol{a}|\boldsymbol{s}) = 0
\tag{33}
$$

$$
h'_f(\frac{\pi(\boldsymbol{a}|\boldsymbol{s})}{\mu(\boldsymbol{a}|\boldsymbol{s})}) + \alpha(\boldsymbol{s}) + \eta \left(Q(\boldsymbol{s}, \boldsymbol{a}) - V(\boldsymbol{s})\right)^2 - \lambda(\boldsymbol{a}|\boldsymbol{s}) = 0
\tag{34}
$$

Since $h'_f$ is a strictly increasing function, its inverse function exists and is also a strictly increasing function. Let $g_f = (h'_f)^{-1}(x)$ be its inverse function. From Equation 34, we can get

$$
\pi(\boldsymbol{a}|\boldsymbol{s}) = \mu(\boldsymbol{a}|\boldsymbol{s})g_f \left(\lambda(\boldsymbol{a}|\boldsymbol{s}) - \alpha(\boldsymbol{s}) - \eta \left(Q(\boldsymbol{s}, \boldsymbol{a}) - V(\boldsymbol{s})\right)^2\right)
\tag{35}
$$

Given a state $\boldsymbol{s}$, we can get $\lambda(\boldsymbol{a}|\boldsymbol{s}) = h'_f(\frac{\pi}{\mu}) + \alpha(\boldsymbol{s}) + \eta \left(Q(\boldsymbol{s}, \boldsymbol{a}) - V(\boldsymbol{s})\right)^2$ from Equation 34, then

(a) If $\lambda(\boldsymbol{a}|\boldsymbol{s}) = h'_f(\frac{\pi}{\mu}) + \alpha(\boldsymbol{s}) + \eta \left(Q(\boldsymbol{s}, \boldsymbol{a}) - V(\boldsymbol{s})\right)^2 > 0$, then $\pi(\boldsymbol{a}|\boldsymbol{s})$ is zero due to complementary slackness. Note that $\pi(\boldsymbol{a}|\boldsymbol{s}) = 0$, thus $h'_f(0) + \alpha(\boldsymbol{s}) + \eta \left(Q(\boldsymbol{s}, \boldsymbol{a}) - V(\boldsymbol{s})\right)^2 > 0$ and we can get $g_f(-\alpha(\boldsymbol{s}) - \eta \left(Q(\boldsymbol{s}, \boldsymbol{a}) - V(\boldsymbol{s})\right)^2) < g_f(h'_f(0)) = 0$.

(b) If $\lambda(\boldsymbol{a}|\boldsymbol{s}) = 0$, then $h'_f(\frac{\pi}{\mu}) + \alpha(\boldsymbol{s}) + \eta \left(Q(\boldsymbol{s}, \boldsymbol{a}) - V(\boldsymbol{s})\right)^2$ is zero and $\pi(\boldsymbol{a}|\boldsymbol{s}) = \mu(\boldsymbol{a}|\boldsymbol{s})g_f \left(-\alpha(\boldsymbol{s}) - \eta \left(Q(\boldsymbol{s}, \boldsymbol{a}) - V(\boldsymbol{s})\right)^2\right) \geq 0$. Note that $\pi(\boldsymbol{a}|\boldsymbol{s}) \geq 0$, thus $h'_f(0) + \alpha(\boldsymbol{s}) + \eta \left(Q(\boldsymbol{s}, \boldsymbol{a}) - V(\boldsymbol{s})\right)^2 \leq 0$ and we can get $g_f(-\alpha(\boldsymbol{s}) - \eta \left(Q(\boldsymbol{s}, \boldsymbol{a}) - V(\boldsymbol{s})\right)^2) \geq g_f(h'_f(0)) = 0$.

Through analysis (a) and (b), we can resolve optimal policy $\pi^*(\boldsymbol{a}|\boldsymbol{s})$ as

$$
\pi^\star(\boldsymbol{a}|\boldsymbol{s}) = \mu(\boldsymbol{a}|\boldsymbol{s}) \max \left\{g_f \left(-\alpha(\boldsymbol{s}) - \eta \left(Q(\boldsymbol{s}, \boldsymbol{a}) - V(\boldsymbol{s})\right)^2\right), 0\right\}.
\tag{36}
$$

let $f(x) = \log x$, then $g_f(x) = \exp(x - 1)) > 0$. Substituting back in Equation 36 with $g_f(x) = \exp(x - 1))$, we can get Equation 15. $\qquad \square$

### A.3 PROOF OF PROPOSITION 5.7

*Proof.* The proof of Proposition 5.7 is based on finding a minimum for the Problem IPF-Soft in a region, and then let the value of Problem IPF-Soft at $\pi^*, V^*$ less than the minimum of Problem IPF-Soft in this region to determine the value of $\eta$. Let $\mathcal{U} = \left\{\pi(\boldsymbol{a}|\boldsymbol{s})|\pi(\boldsymbol{a}|\boldsymbol{s}) \geq 0, \int_{\boldsymbol{a}} \pi(\boldsymbol{a}|\boldsymbol{s})d\boldsymbol{a} = 1\right\}$. Since $\inf_{x,y} g(x, y) = \inf_x \inf_y g(x, y)$ and the constraints about $\pi$ and $V$ in Problem IPF-Soft are

independent, we can reformulate Problem IPF-Soft as

$$\min_{\substack{\pi, V \\ s.t.\pi \in \mathcal{U}}} \mathbb{E}_{\boldsymbol{s} \sim d^\pi(\boldsymbol{s}), \boldsymbol{a} \sim \pi(\boldsymbol{a}|\boldsymbol{s})} \left[ f\left( \frac{\pi(\boldsymbol{a}|\boldsymbol{s})}{\mu(\boldsymbol{a}|\boldsymbol{s})} \right) + \eta \left( Q(\boldsymbol{s}, \boldsymbol{a}) - V(\boldsymbol{s}) \right)^2 \right]$$

$$= \min_{\substack{\pi \\ s.t.\pi \in \mathcal{U}}} \min_V \mathbb{E}_{\boldsymbol{s} \sim d^\pi(\boldsymbol{s}), \boldsymbol{a} \sim \pi(\boldsymbol{a}|\boldsymbol{s})} \left[ f\left( \frac{\pi(\boldsymbol{a}|\boldsymbol{s})}{\mu(\boldsymbol{a}|\boldsymbol{s})} \right) + \eta \left( Q(\boldsymbol{s}, \boldsymbol{a}) - V(\boldsymbol{s}) \right)^2 \right]. \tag{37}$$

For $V$, this is an unconstrained problem. Setting the gradient with respect to $V$ to 0 ($\eta > 0$), we obtain that

$$V(\boldsymbol{s}) = \mathbb{E}_{\boldsymbol{a} \sim \pi(\boldsymbol{a}|\boldsymbol{s})} \left[ Q(\boldsymbol{s}, \boldsymbol{a}) \right]. \tag{38}$$

Substituting back in Equation 37, we can get

$$\min_{\substack{\pi \\ s.t.\pi \in \mathcal{U}}} \min_V \mathbb{E}_{\boldsymbol{s} \sim d^\pi(\boldsymbol{s}), \boldsymbol{a} \sim \pi(\boldsymbol{a}|\boldsymbol{s})} \left[ f\left( \frac{\pi(\boldsymbol{a}|\boldsymbol{s})}{\mu(\boldsymbol{a}|\boldsymbol{s})} \right) + \eta \left( Q(\boldsymbol{s}, \boldsymbol{a}) - V(\boldsymbol{s}) \right)^2 \right]$$

$$= \min_{\substack{\pi \\ s.t.\pi \in \mathcal{V}}} \mathbb{E}_{\boldsymbol{s} \sim d^\pi(\boldsymbol{s}), \boldsymbol{a} \sim \pi(\boldsymbol{a}|\boldsymbol{s})} \left[ f\left( \frac{\pi(\boldsymbol{a}|\boldsymbol{s})}{\mu(\boldsymbol{a}|\boldsymbol{s})} \right) + \eta \left( Q(\boldsymbol{s}, \boldsymbol{a}) - V(\boldsymbol{s}) \right)^2 \right], \tag{39}$$

where $\mathcal{V} = \left\{ \pi(\boldsymbol{a}|\boldsymbol{s}) | \pi(\boldsymbol{a}|\boldsymbol{s}) \in \mathcal{U}, V(\boldsymbol{s}) = \mathbb{E}_{\boldsymbol{a} \sim \pi(\boldsymbol{a}|\boldsymbol{s})} \left[ Q(\boldsymbol{s}, \boldsymbol{a}) \right] \right\}$. Note that $\mathcal{V}$ is the feasible set of Problem IPF and the left-hand side of Equation 39 is exactly Problem IPF.

Let $\mathcal{T} = \left\{ \pi | \pi \in \mathcal{V}, \pi \in \mathring{U}(\pi^*, \sigma) \right\}$, where $\mathring{U}(\pi^*, \sigma) = \left\{ \pi | 0 < |\pi - \pi^*| < \sigma, \sigma > 0 \right\}$. Note that $\mathcal{T} \notin \emptyset$, since $\mathcal{V}$ is a convex set and $\pi^* \in \mathcal{V}$.

Assume Equation 39 can achieve the minimum in $\mathcal{T}$; if it cannot, it indicates a minimum at $\pi^*$ and $V^*$, and Proposition 5.7 holds. We only need to adjust the value of $\eta$ to ensure that the value of Equation 39 at $\pi^*, V^*$ is less than the minimum, thereby proving Proposition 5.7.

$$k^* = \min_{\substack{\pi \\ s.t.\pi \in \mathcal{T}}} \mathbb{E}_{\substack{\boldsymbol{s} \sim d^\pi(\boldsymbol{s}) \\ \boldsymbol{a} \sim \pi(\boldsymbol{a}|\boldsymbol{s})}} \left[ f\left( \frac{\pi(\boldsymbol{a}|\boldsymbol{s})}{\mu(\boldsymbol{a}|\boldsymbol{s})} \right) + \eta \left( Q(\boldsymbol{s}, \boldsymbol{a}) - V(\boldsymbol{s}) \right)^2 \right] \tag{40}$$

Therefore, if the value of Equation 39 at $\pi^*, V^*$ is less than $k^*$, then $\pi^*, V^*$ is a local minimizer of Problem IPF-Soft. Let $h^* = \mathbb{E}_{\substack{\boldsymbol{s} \sim d^\pi(\boldsymbol{s}) \\ \boldsymbol{a} \sim \pi^*(\boldsymbol{a}|\boldsymbol{s})}} \left[ \left( Q(\boldsymbol{s}, \boldsymbol{a}) - V^*(\boldsymbol{s}) \right)^2 \right]$, we can get

$$p^* + \eta h^* = k^*. \tag{41}$$

where $p^*$ is the global solution of problem IPF. Here, for simplicity, we treat $\eta$ as a hyperparameter rather than solving for its exact value. So if $\eta$ satisfies Equation 41, we can get $\pi^*, V^*$ is a local minimizer of Problem IPF.

$\square$

## B    EXTRA RELATED WORK

**Diffusion Model in Offline RL**. Due to our method using the diffusion model for modeling behavior policy, we review works that incorporate the Diffusion model in offline RL. There exist several works that introduce the diffusion model to RL. Diffuser (Janner et al., 2022) uses the diffusion model to directly generate trajectory guided with gradient guidance or reward. DiffusionQL (Wang et al., 2022) uses the diffusion model as an actor and optimizes it through the TD3+BC-style objective with a coefficient $\eta$ to balance the two terms. AdaptDiffuser Liang et al. (2023) uses a diffusion model to generate extra trajectories and a discriminator to select desired data to add to the training set to enhance the adaptability of the diffusion model. DD (Ajay et al., 2022) uses a conditional diffusion model to generate trajectory and compose skills. Unlike Diffuser, DD diffuses only states and trains inverse dynamics to predict actions. QGPO Lu et al. (2023) uses the energy function to guide the sampling process and proves that the proposed CEP training method can get an unbiased estimation of the gradient of the energy function under unlimited model capacity and data samples. SfBC (Chen et al., 2022) first trains a diffusion-based policy and then selects actions based on the $Q$ value, similar

to AWR.IDQL (Hansen-Estruch et al., 2023) reinterpret IQL as an Actor-Critic method and extract the policy through sampling from a diffusion-parameterized behavior policy with weights computed from the IQL-style critic. EDP (Kang et al., 2024) focuses on boosting sampling speed through approximated actions. SRPO (Chen et al., 2023) uses a Gaussian policy in which the gradient is regularized by a pretrained diffusion model to recover the IQL-style policy. Our method is distinct from these methods because we aim to align the implied policy with the value function.

## C    DIFFUSION MODEL

**Diffusion Probabilistic Model (DPM).** Diffusion models (Sohl-Dickstein et al., 2015; Ho et al., 2020; Song & Ermon, 2019) are composed of two processes: the forward diffusion process and the reverse process. In the forward diffusion process, we gradually add Gaussian noise to the data $x_0 \sim q(x_0)$ in $T$ steps. The step sizes are controlled by a variance schedule $\beta_i$:

$$q(x^{1:T} \mid x^0) := \prod_{i=1}^{T} q(x^i \mid x^{i-1}),$$
$$q(x^i \mid x^{i-1}) := \mathcal{N}(x^i; \sqrt{1 - \beta_i} x^{i-1}, \beta_i I). \tag{42}$$

In the reverse process, we can recreate the true sample $x_0$ through $p(x^{i-1}|x^i)$:

$$p(x) = \int p(x^{0:T}) dx^{1:T}$$
$$= \int \mathcal{N}(x^T; 0, I) \prod_{i=1}^{T} p(x^{i-1}|x^i) dx^{1:T}. \tag{43}$$

The training objective is to maximize the ELBO of $\mathbb{E}_{q_{x_0}} [\log p(x_0)]$. Following DDPM (Ho et al., 2020), we use the simplified surrogate loss

$$\mathcal{L}_d(\phi) = \mathbb{E}_{i \sim [1,T], \epsilon \sim \mathcal{N}(0,I), x_0 \sim q} \left[ ||\epsilon - \epsilon_\phi(x_i, i)||^2 \right] \tag{44}$$

to approximate the ELBO. After training, sampling from the diffusion model is equivalent to running the reverse process.

**Conditional DPM.** There are two kinds of conditioning methods: classifier-guided (Dhariwal & Nichol, 2021) and classifier-free (Ho & Salimans, 2021). The former requires training a classifier on noisy data $x_i$ and using gradients $\nabla_x \log f_\Phi(y|x_i)$ to guide the diffusion sample toward the conditioning information $y$. The latter does not train an independent $f_\Phi$ but combines a conditional noise model $\epsilon_\phi(x_i, i, s)$ and an unconditional model $\epsilon_\phi(x_i, i)$ for the noise. The perturbed noise $w\epsilon_\phi(x_i, i) + (w + 1)\epsilon_\phi(x_i, i, s)$ is used to later generate samples. However (Pearce et al., 2023) shows this combination will degrade the policy performance in offline RL. Following (Pearce et al., 2023; Wang et al., 2022) we solely employ a conditional noise model $\epsilon_\phi(x_i, i, s)$ to construct our noise model ($w = 0$).

## D    IMPLICIT DIFFUSION Q-LEARNING (IDQL)

**Implicit Diffusion Q-learning (IDQL)**. To find the implicit policy in the learned value function, IDQL (Hansen-Estruch et al., 2023) generalizes the value loss in Equation 4 with an arbitrary convex loss $U$ on the difference $Q - V$.

$$V^*(s) = \underset{V(s)}{\arg\min} \, \mathbb{E}_{a \sim \mu(a|s)}[U(Q(s, a) - V(s))] = \underset{V(s)}{\arg\min} \, \mathcal{L}_V^U(V(s)). \tag{45}$$

Under some assumptions about $U$, IDQL derives the implicit policy in optimal $V$ defined in Equation 45

$$w(s, a) = \frac{|U'(Q(s, a) - V^*(s))|}{|Q(s, a) - V^*(s)|}, \tag{46}$$

which yields an expression for the implicit actor as $\pi_{\text{imp}}(a|s) \propto \mu(a|s)w(s, a)$. For expectile loss $f(u) = L_2^\tau(u)$ (from Equation 4), the weight of IDQL is

$$w_2^\tau(s, a) = |\tau - \mathbb{1}(Q(s, a) < V_\tau^2(s))|. \tag{47}$$

## E    PSEUDOCODE

**Algorithm 2** AlignIQL Training

1: Initialize behavior policy network $\mu_\phi$, critic networks $Q_\theta, V_\psi$, and target networks $Q_{\hat{\theta}}$, multiplier networks $\alpha_\omega(\boldsymbol{s}), \beta_\chi(\boldsymbol{s})$
2: **for** $t = 1$ to $T$ **do**
3:     Sample from $\mathcal{B} = \{(\boldsymbol{s}_t, \boldsymbol{a}_t, r_t, \boldsymbol{s}_{t+1})\} \sim \mathcal{D}$.
4:     *# Critic updating*
5:     $\psi \leftarrow \psi - \lambda \nabla_\psi \mathcal{L}_V(\psi)$ (Equation 4)
6:     $\theta \leftarrow \theta - \lambda \nabla_\theta \mathcal{L}_Q(\theta)$ (Equation 5)
7:     **if** AlignIQL-hard: **then**
8:         *# Multiplier network updating*
9:         $\omega \leftarrow \omega + \lambda \nabla_\omega \mathcal{L}_M(\omega)$
10:        $\chi \leftarrow \chi + \lambda \nabla_\chi \mathcal{L}_M(\chi)$
11:     **end if**
12:     $\phi \leftarrow \phi - \lambda \nabla_\phi \mathcal{L}_\mu(\phi)$(Equation 44)
13:     *# Target Networks updating*
14:     $\hat{\theta} \leftarrow (1 - \eta)\hat{\theta} + \eta\theta$
15: **end for**

**Algorithm 3** AlignIQL Policy Extraction

1: **Pretraining:** $Q_{\hat{\theta}}, V_\psi, \mu_\phi$, multiplier networks $\alpha_\omega(\boldsymbol{s}), \beta_\chi(\boldsymbol{s})$
2: Samples per state $N, \eta$
3: **while** not done **do**
4:     Get current state $\boldsymbol{s}$
5:     Sample $a_i \sim \mu_\phi(\boldsymbol{a}|\boldsymbol{s})$, $i = 1, \ldots, N$
6:     **if** AlignIQL-hard: **then**
7:         Compute weight $w(\boldsymbol{s}, \boldsymbol{a})$ through Equation 9
8:     **else**
9:         Compute weight $w(\boldsymbol{s}, \boldsymbol{a})$ through Equation 15
10:     **end if**
11:     Normalize: $p_i = \frac{w(s, a_i)}{\sum_j w(s, a_j)}$
12:     Select $a_{\text{taken}}$ with the highest probability according to $p_i$
13: **end while**

The pseudocode for AlignIQL and AlignIQL-hard is presented in Algorithm 2 and Algorithm 3. Note that when we use AlignIQL, the training process is the same as IQL, to implement AlignIQL, we only need to change the weight used in IQL to our AlignIQL's weight. As shown in Algorithm 1.

## F EXPERIMENTAL DETAILS

**MuJoCo Experiments:** Our Policy-based implementation is based on CORL (Tarasov et al., 2022), an Offline Reinforcement Learning library that provides high-quality and easy-to-follow single-file implementations of SOTA ORL algorithms. Following AWR, we clip the weight of AlignIQL using $\max\{0.01, \text{weight}\}$. We sweep $\eta$ since the alignment of policy depends on the value of $\eta$ in different environments.

**AntMaze Experiments:** Our Energy-based implementation is based on IDQL (Hansen-Estruch et al., 2023) and jaxrl repo which uses the JAX framework to implement RL algorithms. All networks are optimized through the Adam (Kingma & Ba, 2014). We clip the multiplier network gradient to prevent gradient explosion due to the exponential term. For the AntMaze

**Algorithm 1** IQL using AlignIQL or AWR

Initialize parameters $\psi, \theta, \hat{\theta}, \phi$.
TD learning (IQL):
**for** each gradient step **do**
    $\psi \leftarrow \psi - \lambda_V \nabla_\psi L_V(\psi)$
    $\theta \leftarrow \theta - \lambda_Q \nabla_\theta L_Q(\theta)$
    $\hat{\theta} \leftarrow (1 - \alpha)\hat{\theta} + \alpha\theta$
**end for**
# Policy extraction (AWR or AlignIQL):
**for** each gradient step **do**
    # Update policy with Equation 6 # AWR
    Update policy with Equation 15 # AlignIQL
**end for**

experiments in Table 1, we fix $\eta = 1$ and sweep $N$ while keeping other hyperparameters consistent with IDQL. We report the best final evaluation average scores of D-AlignIQL and IDQL under different values of $N$. We use quantile loss and Equation 47 for IDQL since the expectile objective is used in IQL. For networks, we follow the default networks and parameters used by IDQL. The policy network uses an LN_Resnet architecture Hansen-Estruch et al. (2023) (Appendix G) with hidden size 256 and $n = 3$. The critic and value networks are 2-layer MLPs with a hidden size of 256 and ReLU activation functions.

**Sparse Rewards Tasks:** For D-AlignIQL, we use $\eta = 1 > 0$ and $\tau = 0.7$ and keep other hyperparameters the same as Table F. For D-AlignIQL-A, we sweep over $N \in \{256, 512, 1024\}$, and $\tau \in \{0.7, 0.9\}$. For IDQL, we sweep over $\tau \in \{0.7, 0.9\}$. The weight of IDQL is $w_2^\tau(s, a) = |\tau - \mathbb{1}(Q(s, a) < V_\tau^2(s))|$. We report the scores of Table 8 by choosing the best score from different $N$.

We provide the main hyperparameters in Table F to reproduce our results in D4RL. Following IDQL, we use normalization to adjust the rewards, which means $r = r/(r_{\max} - r_{\min})$. For AntMaze tasks, $r = r - 1$. We also follow the IDQL's advice to take the maximum probability action at evaluation

time. We train for 300000 epochs for AntMaze tasks with batch size 512 and 100000 epochs for MuJoCo tasks with batch size 256, consistent with IDQL and CORL. Here are the hyperparameters for reproducing our results.

| | |
|---|---|
| **LR** (For all networks except for multiplier) | 3e-4 |
| **LR** ( Multiplier Network ) | 3e-5 |
| **Critic Batch Size** | 512 |
| **Actor Batch Size** | 512 |
| $\tau$ **Expectiles** | 0.7 (locomotion), 0.9 (AntMaze) |
| | 0.5 (Half-ME) |
| | 5 (Half-MR,Hopper-ME,Hopper-MR) |
| $\eta$ **For AlignIQL and D-AlignIQL** | 1 (Half-M,Walker-MR) |
| | 10 (Hopper-M,Waler-ME,Hopper-M) |
| | 1 (D-AlignIQL AntMaze) |
| **Grad norm for multiplier on MuJoCo** | 1.0 ($\alpha$), 0.5 ($\beta$) |
| **Grad norm for multiplier on AntMaze** | 1.0 ($\alpha$), 1 ($\beta$) |
| **Critic Grad Steps** | 3e6 |
| **Actor Grad Steps** | 3e6 |
| **Target Critic EMA** | 0.005 |
| **T** | 5 |
| **Beta schedule** | Variance Preserving (Song et al., 2020) |
| **Actor Dropout Rate** | 0.1 for actor on all tasks |
| **Critic Dropout Rate** | 0.1 for AntMaze Tasks in AlignIQL-hard |
| **Number Residual Blocks** | 3 |
| **Actor Cosine Decay (Loshchilov & Hutter, 2016)** | Number of Actor Grad Steps |
| **Optimizer** | Adam (Kingma & Ba, 2014) |
| | 256 (umaze) |
| | 16 (umaze-d) |
| | 256 (medium-p) |
| **Best $N$ For D-AlignIQL** | 2048 (medium-d) |
| | 64 (large-p) |
| | 64 (large-d) |

## F.1 RESULTS OF ALIGNIQL-HARD

In this chapter, we report the results of D-AlignIQL-hard. Table F reports the hyperparameters we used for AlignIQL-hard.

Table 4: Average Results of D-AlignIQL-hard on AntMaze tasks.

| | D-AlignIQL-hard | | | D-AlignIQL | | |
|---|---|---|---|---|---|---|
| D4RL Tasks | $N = 16$ | $N = 64$ | $N = 256$ | $N = 16$ | $N = 64$ | $N = 256$ |
| **AntMaze** | 54.2 | 57.9 | 56.7 | 65.8 | 70.2 | 70.7 |

We also report the performance of D-AlignIQL under different $N$. Therefore, the results in Table 4 are slightly lower than those in Table 1. For D-AlignIQL and D-AlignIQL-hard, especially for D-AlignIQL-hard, we perform minimal hyperparameter tuning. In most cases, we use the default

parameters of IDQL. Therefore, the performance of our algorithm can be further improved with additional tuning.

## F.2 RUNNING TIME

The biggest problem of the diffusion-based method is the long inference time, which comes from the iterative running of the Markov chain. In this part, we present the running time of D-AlignIQL compared to other methods. We tested the runtime of DiffCPS on an RTX 3050 GPU on D4RL tasks. (3000 epochs (3e6 gradient steps)) From Table 5, it's evident that the runtime of D-AlignIQL is comparable to other diffusion-based methods.

Table 5: Runtime of different diffusion-based offline RL methods. (Average)

| D4RL Tasks | D-AlignIQL (ours) (T=5) | DiffusionQL (T=5) | SfBC (T=5) | IDQL (T=5) |
|---|---|---|---|---|
| **Locomotion Runtime** (1 **epoch**) | 9.12s | 5.1s | 8.4s | 9.5s |
| **AntMaze Runtime** (1 **epoch**) | 9.76s | 10.5s | 10.5s | 10.5s |

Although the runtime of D-AlignIQL is comparable to other diffusion-based methods, AlignIQL is still slower than the Gaussian-based policy (about 1.2s for one epoch). The slow inference speed can harm the performance in real-time robot control tasks. Fortunately, this problem can be solved by recent sample acceleration methods, like SiD (Zhou et al., 2024b;a) or EDP (Kang et al., 2024). EDP directly constructs actions from corrupted ones at training to avoid running the sampling chain. In this way, EDP only needs to run the noise-prediction network once, which can substantially reduce the training time. Below, we first shortly introduce EDP

**EDP:** Kang et al. (2024) noticed that the noisy sample of diffusion model can be written as $q(\mathbf{x}^t|\mathbf{x}^0) = \mathcal{N}(\mathbf{x}^t; \sqrt{\bar{\alpha}_t}\mathbf{x}^0, (1-\bar{\alpha}_t)\mathbf{I})$.

Using the parametrization trick, we can get

$$\boldsymbol{x}^t = \sqrt{\bar{\alpha}_t}\boldsymbol{x}^0 + \sqrt{1-\bar{\alpha}_t}\epsilon, \quad \epsilon \in \mathcal{N}(\mathbf{0}, \mathbf{I}) \tag{48}$$

Replacing $\epsilon$ with our denoising network $\epsilon_\phi(\boldsymbol{x}_i, i, \boldsymbol{s})$, we can obtain the action by running the noise-prediction once:

$$\boldsymbol{x}^0 = \frac{1}{\sqrt{\bar{\alpha}_t}}\boldsymbol{x}^t - \frac{\sqrt{1-\bar{\alpha}_t}}{\sqrt{\bar{\alpha}_t}}\epsilon_\phi(\boldsymbol{x}_i, i, \boldsymbol{s}) \tag{49}$$

Although EDP is a simple method, it can greatly reduce the training time of diffusion-based offline RL methods while keeping competitive results. EDP can also enjoy the benefits of other diffusion acceleration methods, like DPM-solver Lu et al. (2022).

We use the EDP's official IQL code to reimplement our method. In fact, reimplementing AlignIQL based on IQL is very simple, we only need to change one line code corresponding to the policy extraction step as shown below.

```python
def compute_actor_loss(
    self, batch: TorchMiniBatch, action: None
):
    # compute weight
    with torch.no_grad():
        v = self._modules.value_func(batch.observations)
        min_Q = self._targ_q_func_forwarder.compute_target(
            batch.observations, reduction="min"
        ).gather(1, batch.actions.long())
    # Weights for AlignIQL used in extracting the IQL policy
    exp_a = torch.exp(((min_Q - v)**2) * self.eta).clamp(
        max=self._max_weight
    )
    # Weights for AWR used in extracting the IQL policy
    # exp_a = torch.exp((min_Q - v) * self._weight_temp).clamp(
    #     max=self._max_weight
    #)
```

```
            # compute log probability
            dist = self._modules.policy(batch.observations)
            log_probs = dist.log_prob(batch.actions.squeeze(-1)).unsqueeze(1)

            return ActorLoss(-(exp_a * log_probs).mean())
```

Table 6 shows the results of EDP-based AlignIQL.

Table 6: Performance and runtime time (1 epoch) of D-AlignIQL (Diffusion steps $T = 5$) and EDP-based D-AlignIQL.

| Method | Performance | | Runtime (s) | |
|---|---|---|---|---|
| | Large-p | Large-d | Large-p | Large-d |
| D-AlignIQL | 65.2 | 66.4 | 9.5 | 9.78 |
| EDP-based D-AlignIQL | 43 | 62 | 2.22 | 1.95 |

The above results are conducted on a 1 random seed since we mainly focus on the runtime. Table 6 shows that simple EDP-based AlignIQL can reduce at most $80\%$ training time while matching the performance of policy with origin diffusion-based policy. Note that we do not use theDPM-solver in our code, which can add an additional $2.3x$ training speedup according to EDP's origin paper. In brief, the diffusion-based policy with sample acceleration can match the speed of the Gaussian policy (about $1.2$s for one epoch).

### F.3 EXTRA ABLATION STUDY

Table 7 shows the best results of different $\eta$ on selected tasks.

Table 7: Performance of AlignIQL under different $\eta$

| $\eta$ | Walker2d | | Halfcheetah | |
|---|---|---|---|---|
| | ME | MR | ME | MR |
| $\eta = 3$ | 110.3 | 77.4 | 82.1 | 42.6 |
| $\eta = 5$ | 110.4 | 79.5 | 81.4 | 42.7 |
| $\eta = 10$ | 110.5 | 80.1 | 80.1 | 42.5 |

## G SPARSE REWARD TASKS

**Results on D4RL Sparse Reward Tasks**. We conduct the experiment on Adroit tasks. Compared with MuJoCo tasks, the Adroit tasks are high dimensional and feature sparse rewards. For human and expert datasets, the data is collected from human demonstrators. Results can be found in Table 8. Table 8 shows that compared to IDQL, AlignIQL can achieve competitive results and significantly outperform IDQL on some challenging tasks like relocate-human.

## H DISCUSSION ON DIFFERENT REGULARIZERS

In this chapter, we aim to validate the effect of different regularizers. We experimented with the case of $f(x) = x - 1$ in D-AlignIQL-hard and D-AlignIQL. Let $f(x) = x - 1$, we can get $g_f(x) = \frac{1}{2}x + \frac{1}{2}$. Substituting back in Equation 36 and Equation 9 with $g_f(x) = \frac{1}{2}x + \frac{1}{2}$, we can get

$$\text{AlignIQL:} \quad \pi^\star(\boldsymbol{a}|\boldsymbol{s}) = \mu(\boldsymbol{a}|\boldsymbol{s}) \max\left\{ \frac{1}{2}\left(-\alpha(\boldsymbol{s}) - \eta\left(Q(\boldsymbol{s},\boldsymbol{a}) - V(\boldsymbol{s})\right)^2\right) + \frac{1}{2}, 0 \right\}, \quad (50)$$

$$\text{AlignIQL-hard:} \quad \pi^\star(\boldsymbol{a}|\boldsymbol{s}) = \mu(\boldsymbol{a}|\boldsymbol{s}) \max\left\{ \frac{1}{2}\left(-\alpha(\boldsymbol{s}) - \beta(\boldsymbol{s})Q(\boldsymbol{s},\boldsymbol{a})\right) + \frac{1}{2}, 0 \right\}. \quad (51)$$

Table 8: Normalized scores of D-AlignIQL against other baselines on D4RL sparse-reward tasks. We **bold** the mean values that $\geq 0.99 *$ highest value. "-A" indicates that we sweep over all the hyperparameters. D-AlignIQL refers to fixed $\eta = 1$ and $\tau = 0.7$.

| Task | BC | BCQ | CQL | IQL | D-AlignIQL-A | Algae-DICE | IDQL | D-AlignIQL |
|------|-----|-----|-----|-----|--------------|------------|------|------------|
| pen-human | 63.9 | 68.9 | 37.5 | 71.5 | **76.0**±4.8 | -3.3 | 70.7±8.4 | **76.0**±4.8 |
| hammer-human | 1.2 | 0.5 | **4.4** | 1.4 | 2.25±0.01 | 0.3 | 2.8±0.8 | 2.0±0.7 |
| door-human | 2.0 | 0.0 | **9.9** | 4.3 | 6.0±3.6 | 0.0 | 5.2±1.3 | 6.0±3.6 |
| relocate-human | 0.1 | -0.1 | 0.2 | 0.1 | **0.67**±0.14 | -0.1 | 0.09±0.02 | 0.28±0.34 |
| pen-expert | 85.1 | 114.9 | 107.0 | 111.7 | 127.3±1.2 | -3.5 | **132.9**±4.5 | 116.0±4.3 |
| hammer-expert | **125.6** | 107.2 | 86.7 | 116.3 | 125.5±0.25 | 0.3 | **126.5**±0.7 | 124.7±1.9 |
| door-expert | 34.9 | 99.0 | 101.5 | 103.8 | **105.0**±0.5 | 0.0 | **105.0**±0.1 | **104.6**±0.5 |
| relocate-expert | 101.3 | 41.6 | 95.0 | 102.7 | **108.3**±0.2 | -0.1 | **108.3**±1.3 | 106.0±1.5 |

We conducted experiments on Antmaze-umaze to evaluate the effects of different regularizers. We keep all other hyperparameters the same as Table F. The experimental details are described as follows.

**AlignIQL:** In Equation 50, $\alpha(\boldsymbol{s})$ serves as a normalization term, which does not affect the action evaluation when $\frac{1}{2}\left(-\alpha(\boldsymbol{s}) - \eta\left(Q(\boldsymbol{s}, \boldsymbol{a}) - V(\boldsymbol{s})\right)^2\right) + \frac{1}{2} \geq 0$. To simply the training process, we assume $\frac{1}{2}\left(-\alpha(\boldsymbol{s}) - \eta\left(Q(\boldsymbol{s}, \boldsymbol{a}) - V(\boldsymbol{s})\right)^2\right) + \frac{1}{2} \geq 0$ and ignore $\alpha(\boldsymbol{s})$. Since we use the energy-based implementation and select the action with maximum weight, such simplification is reasonable and avoids training an extra multiplier network. We set $\eta = 1$ in the Antmaze umaze experiment.

**AlignIQL-hard:** Similar to Lemma 5.3, we can train our multiplier through the following loss function (we replace $\frac{1}{2}\left(-\alpha(\boldsymbol{s}) - \beta(\boldsymbol{s})Q(\boldsymbol{s}, \boldsymbol{a})\right) + \frac{1}{2}$ with $w_{\text{linear}}$ for simplicity)

$$\min_{\alpha, \beta} \mathcal{L}_M = \mathbb{E}_{\boldsymbol{a} \sim \mu}\left[\mathbb{1}\left(w_{\text{linear}} > 0\right) w_{\text{linear}}^2\right] + \alpha(\boldsymbol{s}) + \beta(\boldsymbol{s})V(\boldsymbol{s}), \tag{52}$$

*Proof.* This proof can be obtained by setting the gradient of Equation 52 to 0 with respect to $\alpha, \beta$. □

Table 9: Performance of different regularizers in D-AlignIQL and D-AlignIQL-hard

| Regularizers | D-AlignIQL | | D-AlignIQL-hard | |
|--------------|------------|---------|-----------------|---------|
| | umaze-p | umaze-d | umaze-p | umaze-d |
| $f(x) = \log x$ | 94.8 | 82.4 | 84.7 | 74.0 |
| $f(x) = x - 1$ | 95.0 | 87.0 | 92.0 | 70.0 |

The results of $f(x) = x - 1$ in Table 9 is evaluated over 2 random seed. We found that the performance of the linear regularizer is comparable to the results of D-AlignIQL in Table 9. This is because both place more weight on actions with higher $-(Q(\boldsymbol{s}, \boldsymbol{a}) - V(\boldsymbol{s}))^2$. For $f(x) = x - 1$ in D-AlignIQL-hard, we found that it like $f(x) = \log x$, is susceptible to hyperparameters, especially the learning rate of the Lagrange multiplier network, and both showed a certain decline in performance by the end of training. We attribute this performance drop to the susceptibility of the multiplier network to hyperparameters, and future improvements to the multiplier network and hyperparameters may address this issue.

## I  NOISE DATA EXPERIMENT

In this chapter, we follow the Yang et al. (2023) to test the robustness of our method. Specifically, we evaluate the performance of our method across diverse data corruption scenarios, including random attacks on states, actions, rewards, and next-states. The random corruption is applied by adding random noise to the attacked elements of a $c$ portion of the datasets. The corruption scale is controlled by $\epsilon$. The details of data corruption can be found in Yang et al. (2023) (Appendix D). Here we briefly introduce the random data corruption on states, actions, rewards, and next-states.

**Random observation attack:** $\hat{s} = s + \lambda \cdot \text{std}(s)$. $\lambda \sim \text{Uniform}\left[-\epsilon, \epsilon\right]$

**Random action attack:** $\hat{a} = a + \lambda \cdot \text{std}(a)$. $\lambda \sim \text{Uniform}\left[-\epsilon, \epsilon\right]$.

**Random reward attack:** $\hat{r} \sim \text{Uniform}\left[-30 \cdot \epsilon, 30 \cdot \epsilon\right]$.

**Random dynamics attack:** $\hat{s}' = s' + \lambda \cdot \text{std}(s')$. $\lambda \sim \text{Uniform}\left[-\epsilon, \epsilon\right]$.

We train for $2e6$ steps on the D4RL halfcheetah-medium-replay-v2 robust tasks with $\epsilon = c = 0.5$. Note that we use policy-based AlignIQL (Algorithm 1) in robust experiments (Appendix I) and image-based control (Appendix J]), which means employing a Gaussian-based policy instead of the diffusion model. The reason for using a Gaussian-based policy is to prove our method can be generalized to any type of policy. We reimplement our method based on the official code from Yang et al. (2023), a robust version of IQL with Q-ensemble and Huber regression. As shown in Appendix F.2, implementing our code based on IQL is very straightforward, requiring only changes to the policy extraction step. We use $\beta = \eta = 3$ for both IQL+AWR (Abbreviated as IQL) and AlignIQL. For $\tau$, we adopt the default value $\tau = 0.7$ provided in the official code from Yang et al. (2023). We report the 5-Running average at step $2e6$. Our method, AlignIQL, achieves the highest average

Table 10: Results of Robust Experiment in Halfcheetah-medium-replay-v2 over 3 random seeds. AlignIQL outperforms IQL a lot under observation attack.

| Method | Halfcheetah | | | | |
| | Reward | Action | Dynamics | Observation | Average |
|---|---|---|---|---|---|
| **AlignIQL** | 40.2 | 40.23 | 37.20 | **29.05** | **36.50** |
| **IQL** | 42.15 | 39.47 | **37.40** | 23.14 | 35.54 |
| **CQL** | **43.56** | 44.76 | 0.06 | 28.51 | 29.22 |

scores compared to other methods. More importantly, AlignIQL demonstrates greater robustness against observation attacks compared to IQL. While CQL performs well under attacks on actions, observations, and rewards, it fails to learn under dynamics attacks. Since policy alignment relies on the value function, the performance of AlignIQL may degrade under reward attacks. However, AlignIQL demonstrates greater robustness against observation attacks, as it assigns higher weights to actions for which $Q$ approaches $V(s)$. $V(s)$ is learned by a neural network, which exhibits robustness against corrupted inputs (e.g., corrupted observations) because similar states tend to have similar $V(s)$. However, in the context of RL, $Q(s, a)$ may vary significantly across similar states. This may explain why our method performs better under observation attacks.

## J    VISION-BASED CONTROL

In this chapter, we report the results of AlignIQL and IQL on the Atari tasks (Agarwal et al., 2020). Specifically, we choose three image-based Atari games with discrete action spaces: Breakout, Qbert, and Seaquest. We use d3rlpy, a modularized offline RL library that includes several SOTA offline RL algorithms and offers an easy-to-use wrapper for the offline Atari datasets introduced by Agarwal et al. (2021). To increase the task difficulty, we use only $1\%$ or $0.5\%$ of the transitions from all epochs in the original datasets. ($1\text{M} \times 50\text{epoch} \times 1\% \quad \text{or} \quad 0.5\%$)

We implement the discrete version of AlignIQL (D-AlignIQL) based on the discrete IQL (D-IQL) from d3rlpy. As shown in Appendix F.2, there is no price to implement AlignIQL based on IQL. For D-IQL+AWR (Abbreviated as IQL), we report the average score of the last 3 evaluations by selecting the minimal standard deviation from $\tau \in [0.5, 0.7, 0.9]$ in the last 3 evaluations. Similarly, for D-AlignIQL with $\eta = 1$, we report the average score of the last 3 evaluations by selecting the minimal standard deviation from $\tau \in [0.5, 0.7, 0.9]$ in the last 3 evaluations. We do this because vision-based methods are unstable, and their performance may fluctuate significantly across different seeds or training steps.

The only difference between AlignIQL and IQL lies in the method of extracting policies. AlignIQL achieves the best performance in 5 out of 6 games and exhibits a smaller standard deviation compared to IQL. We also observed that in certain tasks, AlignIQL or IQL performs better on smaller datasets.

This phenomenon was also observed when training CQL on Atari tasks, as reported in Xu et al. (2023).

Table 11: Performance in setting with $1\%$ or $0.5\%$ Atari dataset. AlignIQL achieves the best performance in 5 out of 6 games.

| Method | Breakout | | Qbert | | Seaquest | |
|---|---|---|---|---|---|---|
| | $1\%$ | $0.5\%$ | $1\%$ | $0.5\%$ | $1\%$ | $0.5\%$ |
| AlignIQL | $\mathbf{9.23} \pm 0.8$ | $\mathbf{7.13} \pm 2.5$ | $\mathbf{7170} \pm 877$ | $\mathbf{7512} \pm 548$ | $192.7 \pm 30.02$ | $\mathbf{371.3} \pm 1.1$ |
| IQL | $6.87 \pm 1.1$ | $5.3 \pm 3.2$ | $4160 \pm 1473$ | $3773.3 \pm 780.2$ | $\mathbf{238.7} \pm 21.6$ | $306.7 \pm 25.2$ |

## K    TRAINING CURVES OF ANTMAZE TASKS

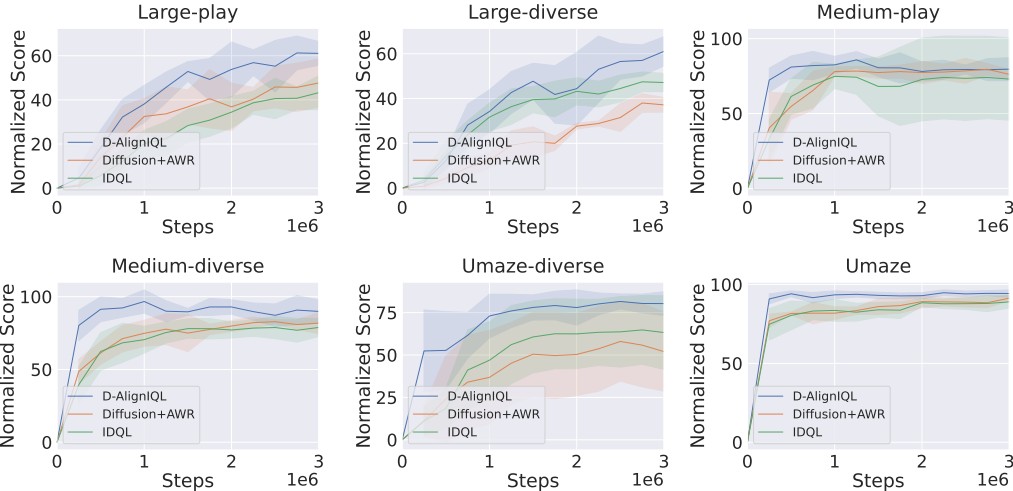

Figure 3: Training curves of D-AlignIQL, IDQL, and Diffusion+AWR. The normalized score is calculated by averaging the scores across three different $N$ ($N = 16, 64, 256$) except for Medium tasks, where $N$ represents the number of actions generated by the diffusion-based behavior policy. For Medium tasks, the normalized score is calculated by averaging the scores for $N$ ($N = 256, 1024, 2048$).

