# OpenReview forum: "AlignIQL: Policy Alignment in Implicit Q-Learning through Constrained Optimization"
_ICLR.cc/2025/Conference — Submitted to ICLR 2025_

### Official Review · Reviewer_kMNe · 2024-10-24

**Soundness:** 3
**Presentation:** 3
**Contribution:** 3
**Rating:** 5
**Confidence:** 3

**Summary:**

The paper introduces AlignIQL, a novel approach to extracting implicit policies in offline reinforcement learning by formulating the implicit policy-finding problem as a constrained optimization problem. AlignIQL and its variant AlignIQL-hard leverage policy alignment constraints to ensure that the extracted policy reflects the learned Q-function while maintaining the advantages of decoupling the actor and critic in Implicit Q-Learning (IQL). The authors demonstrate that their method achieves competitive or superior performance on D4RL datasets, particularly excelling in complex sparse reward tasks like AntMaze and Adroit, while also being more robust to hyperparameter variations than existing methods. Additionally, the study provides theoretical insights into the conditions under which weighted regression can be effectively utilized for policy extraction in IQL. Overall, the proposed algorithms contribute to a better understanding of the bottlenecks in IQL-style methods and offer a more effective means for implicit policy extraction in offline reinforcement learning.

**Strengths:**

- The introduction of AlignIQL as a constrained optimization approach represents a significant advancement in offline reinforcement learning, providing a fresh perspective on implicit policy extraction.
- The empirical results demonstrate that AlignIQL and its variant achieve competitive performance across a variety of D4RL benchmarks, particularly in challenging tasks with sparse rewards, indicating the effectiveness of the proposed methods.
- Theoretical Insights: The paper offers valuable theoretical analysis regarding the use of weighted regression for policy extraction, enhancing the understanding of the underlying mechanisms that contribute to the success of IQL methods.
- By incorporating policy alignment constraints, the approach ensures that the extracted policies are both effective and representative of the learned Q-function, leading to improved stability and reliability in offline settings.
- AlignIQL shows increased robustness to variations in hyperparameters compared to existing methods, which is crucial for practical applications where tuning can be challenging.

**Weaknesses:**

- While the experiments demonstrate competitive performance on specific D4RL benchmarks, the applicability of AlignIQL to other domains or more diverse environments may not be fully established, limiting its generalizability.
- The proposed framework may introduce additional complexity in implementation compared to existing methods, which could deter practitioners who seek simpler solutions for offline reinforcement learning.
- Although the paper includes comparisons with several baseline methods, it may benefit from a more comprehensive analysis against a broader range of state-of-the-art algorithms to fully contextualize its contributions.
- The performance improvements may be contingent on the quality of the dataset used, raising concerns about the approach’s robustness in real-world scenarios where data can be noisy or incomplete.
- The computational requirements for training AlignIQL could be higher than those of simpler methods, potentially limiting its scalability for larger-scale applications or real-time scenarios.

**Questions:**

- How does AlignIQL perform in real-world environments with noisy or incomplete datasets? The paper evaluates performance on D4RL benchmarks, but it would be interesting to see how the method handles imperfect data.
- What are the key factors that affect the alignment between the Q-values and the learned policy in AlignIQL? Understanding the sensitivity of the method to different alignment parameters could help clarify its robustness.
- How does the computational complexity of AlignIQL compare to other state-of-the-art offline reinforcement learning methods in terms of training time and resource usage? This would help evaluate the method’s scalability for larger or more complex tasks.
- Is the approach compatible with more advanced neural network architectures, such as transformers, for offline reinforcement learning? Could integrating more modern architectures improve its performance?
- What are the potential limitations of applying AlignIQL to tasks outside of continuous control, such as discrete action spaces or hierarchical tasks? This could provide insight into the method’s broader applicability.

---

> ### Author Response · Authors · 2024-11-23
>
> We thank reviewer kMNe for the comments and suggestions. Below, we address the concerns raised in your review point by point.
>
> > Q1 The applicability of AlignIQL to other domains or more diverse environments may not be fully established, limiting its generalizability.
>
> We have conducted experiments on different regularizers, vision-based discrete control, and robustness to demonstrate the generalizability of our method, as shown in Appendix H–J.
>
> > Q2 The proposed framework may introduce additional complexity in implementation compared to existing methods, which could deter practitioners who seek simpler solutions for offline reinforcement learning.
>
> Although AlignIQL-hard introduces additional complexity, such as the multiplier network, AlignIQL remains a highly efficient and simple method.  There are two ways to utilize our methods in offline RL (corresponding to Algorithm 1 and Algorithm 3.
> 1. Energy-based implementation: We first use the learned diffusion-based behavior model $\mu_\phi(a|s)$ to generate $N$ action samples. These actions are then evaluated using weights from Eq 15.
>   2. Policy-based implementation: We use Eq 16 to train the policy, which needs the exact probability density of the current policy (Algorithm 1).
>
> In summary, the first method can be used when employing diffusion-based policies, as the probability density of diffusion models is unknown. The second method is applicable when using Gaussian-based policies.
> For the energy-based implementation (D-AlignIQL in the revised paper), we only need to tune $N$, which represents the number of actions generated by the diffusion-based behavior policy. For the policy-based implementation (AlignIQL in the revised paper), reimplementing AlignIQL based on IQL is very straightforward—we only need to modify one line of code corresponding to the policy extraction step, as shown in Appendix F.2.
>
> Overall, compared to other methods, AlignIQL is as simple as IQL, as it is essentially a policy extraction method for IQL.
>
> > Q3 concerns about the approach’s robustness in real-world scenarios where data can be noisy or incomplete.
>
> We have conducted experiments to evaluate the robustness of our method, based on Yang [1], as detailed in Appendix I. Specifically, we assessed the performance of our method across various data corruption scenarios, including random attacks on states, actions, rewards, and next states. Below are the results.
>
> Results of Robust Experiment in Halfcheetah-medium-replay-v2.
> The results are averaged over 3 random seeds. AlignIQL outperforms IQL significantly under observation attacks.
> | **Method** | **Reward** | **Action** | **Dynamics** | **Observation** | **Average** |
> | --- | --- | --- | --- | --- | --- |
> | **AlignIQL** | 40.2 | 40.23 | 37.20 | **29.05** | **36.50** |
> | **IQL** | 42.15 | 39.47 | **37.40** | 23.14 | 35.54 |
> | **CQL** | **43.56** | **44.76** | 0.06 | 28.51 | 29.22 |
>
> Our method, AlignIQL, achieves the highest average scores compared to other methods. More importantly, AlignIQL demonstrates greater robustness against observation attacks compared to IQL. While CQL performs well under attacks on actions, observations, and rewards, it fails to learn under dynamics attacks. Since policy alignment relies on the value function, the performance of AlignIQL may degrade under reward attacks. However, AlignIQL demonstrates greater robustness against observation attacks, as it assigns higher weights to actions for which $ Q $ approaches $ V(s) $. $ V(s) $ is learned by a neural network, which exhibits robustness against corrupted inputs (e.g., corrupted observations) because similar states tend to have similar $V(s)$. However, in the context of RL, $ Q(s, a) $ may vary significantly across similar states. This may explain why our method performs better under observation attacks.

---

> ### Author Response · Authors · 2024-11-23
>
> > Q4 The computational requirements for training AlignIQL could be higher than those of simpler methods, potentially limiting its scalability for larger-scale applications or real-time scenarios.
>
> We have conducted a detailed experiment on the runtime of our method, as presented in Appendix F.2. From the table below, it is evident that the runtime of D-AlignIQL is comparable to other diffusion-based methods.
>
> Runtime of Different Diffusion-based Offline RL Methods (Average)
>
> | **D4RL Tasks** | **D-AlignIQL (T=5)** | **DiffusionQL (T=5)** | **SfBC (T=5)** | **IDQL (T=5)** |
> | --- | --- | --- | --- | --- |
> | **Locomotion Runtime (1 epoch)** | 9.12s | 5.1s | 8.4s | 9.5s |
> | **AntMaze Runtime (1 epoch)** | 9.76s | 10.5s | 10.5s | 10.5s |
>
> More importantly, our method can be combined with diffusion-based acceleration methods, such as EDP [2], to substantially reduce training time, as shown in the table below. EDP directly constructs actions from corrupted ones during training, avoiding the need to run the sampling chain. By doing so, EDP only requires running the noise-prediction network once, which significantly reduces the training time.
>
> | **Method** | **Performance (Large-p)** | **Performance (Large-d)** | **Runtime (s, Large-p)** | **Runtime (s, Large-d)** |
> | --- | --- | --- | --- | --- |
> | **D-AlignIQL** | 65.2 | 66.4 | 9.5 | 9.78 |
> | **EDP-based D-AlignIQL** | 43 | 62 | 2.22 | 1.95 |
>
>
> The above results are conducted on a $ 1 $ random seed since we mainly focus on the runtime. The above Table shows that simple EDP-based AlignIQL can reduce at most $ 80\% $ training time while matching the performance of policy with origin diffusion-based policy. Note that we do not use the DPM-solver in our code, which can add an additional $2.3x$ training speedup according to EDP's origin paper. In brief, the diffusion-based policy with sample acceleration can match the speed of the Gaussian policy (about $ 1.2 $s for one epoch).
>
> >Q5 What are the key factors that affect the alignment between the Q-values and the learned policy in AlignIQL? Understanding the sensitivity of the method to different alignment parameters could help clarify its robustness.
>
> The key factor affecting alignment is $\eta$. In AlignIQL, $\eta$ is analogous to $\alpha$ in AWR, serving to balance policy alignment with behavior cloning. In fact, $\eta$ can also be interpreted as implicit critic exploitation. Actually, for IQL, the expectile loss approximates the maximum of
>     $Q_{\hat{\theta}}(s,a)$ when $\tau\approx 1$.  We can approximately think $V(s)=\arg\max_{a\sim\mathcal{D}} Q(s,a)$, and thus, according to Eq 15, $\hat{a} = \arg\max_a Q(s, a)$ has a weight of 1, while other actions are weighted by
> $\exp -\eta (Q(s, a) - V(s))^2$.  For a fixed $\eta$, the weights for other actions are smaller than $\arg\max_{a\sim\mathcal{D}} Q(s,a)$. Therefore, Eq 15 approximately recovers the implicit policy $\pi^*(a|s)=\arg\max_{a\sim\mathcal{D}} Q^*(s,a)$ from IQL learned value functions.
>
> > Q6 Is the approach compatible with more advanced neural network architectures, such as transformers, for offline reinforcement learning? Could integrating more modern architectures improve its performance?
>
> Our method is compatible with more advanced neural network architectures, such as transformers. However, combining our method with such architectures may not improve performance in D4RL tasks, as these tasks do not involve complex representation learning, such as processing high-dimensional images or natural language—areas where advanced architectures like transformers excel. Nevertheless, as a policy extraction method, our approach can serve as a component in more complex tasks to enhance performance.
>
> > Q7 What are the potential limitations of applying AlignIQL to tasks outside of continuous control, such as discrete action spaces or hierarchical tasks? This could provide insight into the method’s broader applicability.
>
> We have conducted experiments on vision-based discrete Atari tasks, as detailed in Appendix J. Below are the results.
> Performance in Settings with (1%) or (0.5%) Atari Dataset
> Performance in Settings with (1%) or (0.5%) Atari Dataset
>
> | **Method** | **Breakout (1%)** | **Breakout (0.5%)** | **Qbert (1%)** | **Qbert (0.5%)** | **Seaquest (1%)** | **Seaquest (0.5%)** |
> | --- | --- | --- | --- | --- | --- | --- |
> | **AlignIQL** | **9.23 ± 0.8** | **7.13 ± 2.5** | **7170 ± 877** | **7512 ± 548** | 192.7 ± 30.02 | **371.3 ± 1.1** |
> | **IQL** | 6.87 ± 1.1 | 5.3 ± 3.2 | 4160 ± 1473 | 3773.3 ± 780.2 | **238.7 ± 21.6** | 306.7 ± 25.2|
>
> AlignIQL achieves the best performance in 5 out of 6 games.
>
> [1] Rui Yang, Han Zhong, Jiawei Xu, Amy Zhang, Chongjie Zhang, Lei Han, and Tong Zhang. Towards robust offline reinforcement learning under diverse data corruption. arXiv preprint arXiv:2310.12955, 2023.
>
> [2] Kang, Bingyi, et al. Efficient Diffusion Policies for Offline Reinforcement Learning.

---

> ### Author Response · Authors · 2024-11-26
>
> Dear reviewer,
>
> Thank you once again for investing your valuable time in providing feedback on our paper. Your insightful suggestions have led to significant improvements in our work, and we look forward to possibly receiving more feedback from you. Since the discussion period between the author and reviewer is rapidly approaching its end, we kindly request you to review our responses to ensure that we have addressed all of your concerns. Also, we remain eager to engage in further discussion about any additional questions you may have.
>
> Best,
>
> Authors

---

> > ### Author Response · Authors · 2024-12-03
> >
> > Dear reviewer,
> >
> > Since the discussion period between the author and reviewer is rapidly approaching its end, we were wondering if our response and revision have resolved your concerns. In our responses, we focus on clarifying the robustness, generalizability, simplicity, and computational requirements of the proposed method and provide extensive experiments. If our response has addressed your concerns, we would be grateful if you could re-evaluate our work.
> >
> > Best regards,
> >
> > The Authors

---

### Official Review · Reviewer_pv7F · 2024-10-31

**Soundness:** 2
**Presentation:** 2
**Contribution:** 2
**Rating:** 5
**Confidence:** 3

**Summary:**

This paper proposes AlignIQL (in two versions) to address the implicit policy-finding problem. The authors formulate it as a constrained optimization problem and derive a closed-form solution. The performance of AlignIQL is competitive compared to the baselines.

**Strengths:**

- This paper introduces a new approach to tackle the implicit policy-finding problem, combining theoretical rigor with practical effectiveness in offline RL.
- The proposed algorithm, AlignIQL, performs well across varied tasks, demonstrating versatility and effectiveness across different offline RL benchmarks.

**Weaknesses:**

- While AlignIQL is rigorous, it adds complexity to training by requiring additional multiplier networks and diffusion models, which may increase computational costs and sensitivity to hyperparameters. The scalability of the method is also a concern; can it be extended to image-based tasks?
- The authors do not explain the use of diffusion modeling in the methods section.
- The performance of AlignIQL raises some concerns:
    - The authors argue that MuJoCo tasks are already saturated for offline RL, which I agree with. However, AlignIQL's performance is also considerably worse than Diffusion QL and even worse than IQL in 4 out of 9 tasks. Given that AlignIQL consumes more computational resources, this discrepancy is problematic.
    - There is a significant performance difference between the authors' version and the original IDQL paper, which further leaves the reader uncertain about the supposed improvements in AlignIQL's performance.
    - The results were obtained using inconsistent hyperparameters, yet the authors under-analyze the ablation study and hyperparameter sensitivity.
    - The authors state, “Figure 2 shows that as training time increases, the performance of AlignIQL with different N converges to the same value, which shows that AlignIQL is insensitive to N.” This conclusion is not obvious from Figure 2. A clearer approach would be to report the mean and standard deviation of these scores.
    - The optimization techniques may risk overfitting in AntMaze environments with sparse rewards, potentially reducing generalization to new scenarios. More testing on sparse reward tasks would benefit this submission.
- In this submission, "policy alignment" is defined differently from its use in language models. A more formal definition of “policy alignment” should be provided, or the authors could consider renaming it.
- A minor issue: multiple duplicate references appear in the bibliography (e.g., lines 568-573). Additionally, lines 916-917 may contain editing errors.

**Questions:**

See Weaknesses

---

> ### Author Response · Authors · 2024-11-23
>
> We thank Reviewer pv7F for the comments and suggestions. Below, we address the concerns raised in your review point by point.
>
> > Q1: Increasing Computational Costs and Sensitivity to Hyperparameters
>
> Although AlignIQL-hard requires training an additional multiplier network, AlignIQL remains an efficient and straightforward method. There are two ways to utilize our methods in offline RL (corresponding to Algorithm 1 and Algorithm 3):
>
> 1. **Energy-based implementation**:
>    We first use the learned diffusion-based behavior model $ \mu_\phi(a|s) $ to generate $ N $ action samples. These actions are then evaluated using weights from Eq. (15).
>
> 2. **Policy-based implementation**:
>    We use Eq. (16) to train the policy, which requires the exact probability density of the current policy (Algorithm 1).
>
> In summary, the first method can be used when employing diffusion-based policies, as the probability density of diffusion models is unknown. The second method is applicable when using Gaussian-based policies.
>
> For energy-based implementation (D-AlignIQL in the revised paper), we only need to sweep the $N$, which represents the number of actions generated by the diffusion-based behavior policy. For policy-based implementation, (AlignIQL in the revised paper), reimplementing AlignIQL based on IQL is very simple, we only need to change one line code corresponding to the policy extraction step as shown in Appendix F.2.Except for $\tau$ in IQL, we only need to tune $\eta$ for AlignIQL. Overall, as a method designed to extract policies from the value function learned by IQL, AlignIQL is effective. As we demonstrate below, our method is not sensitive to hyperparameters. (see Q6)
>
> > Q2: Can It Be Extended to Image-Based Tasks?
>
> We conducted experiments based on Atari tasks. The results are provided in Appendix J of the revised paper:
>
> **Performance in Settings with (1%) or (0.5%) Atari Dataset**
>
> | **Method**    | **Breakout (1%)** | **Breakout (0.5%)** | **Qbert (1%)** | **Qbert (0.5%)** | **Seaquest (1%)** | **Seaquest (0.5%)** |
> |---------------|-------------------|---------------------|----------------|------------------|-------------------|---------------------|
> | **AlignIQL**  | **9.23 ± 0.8**    | **7.13 ± 2.5**      | **7170 ± 877** | **7512 ± 548**   | 192.7 ± 30.02     | **371.3 ± 1.1**     |
> | **IQL**       | 6.87 ± 1.1        | 5.3 ± 3.2           | 4160 ± 1473    | 3773.3 ± 780.2   | **238.7 ± 21.6**  | 306.7 ± 25.2        |
>
> AlignIQL achieves the best performance in 5 out of 6 games.
>
> > Q3: The Authors Do Not Explain the Use of Diffusion Modeling in the Methods Section.
>
> Thank you for pointing this out. The reason we did not explain the use of diffusion modeling in the methods section is to emphasize our key contribution—a new method to extract policies from IQL-learned value functions. We have added comments about diffusion models in our methods section in the revised paper.
>
>
> > Q4: AlignIQL's Performance Is Worse Than Diffusion QL and Even Worse Than IQL in Some MuJoCo Tasks.
>
> Although Diffusion QL performs better on the MuJoCo tasks, our method outperforms it on the more complex AntMaze tasks. We have rerun the policy-based AlignIQL (abbreviated as AlignIQL). This version does not rely on a diffusion model and achieves better results, as shown in Table 1 of the revised paper. This may be because, in policy-based AlignIQL, we can adjust $ \eta $ to obtain better results. In the diffusion setting, the hyperparameter $ N $ has a greater influence on performance than $ \eta $, as a higher $ N $ increases the likelihood of finding the “lucky” action that satisfies $ \hat{a} = \arg\max_a Q(s, a) $, and $\eta$ is multiplied across all evaluated actions.
>
> Since MuJoCo is a relatively simple task, the regularization introduced by the diffusion model (BC constraint) and IQL may be overly restrictive, hindering performance improvements.
>
>
> > Q5: Performance Difference Between the Authors' Version and the Original IDQL Paper
>
> In the original IDQL paper, the best average scores on AntMaze and MuJoCo are 79.1 and 82.1, respectively, while in our version, they are 74.4 and 78.0. The results for IDQL in our paper were obtained by running the official code with default hyperparameters. Performance differences in MuJoCo may be caused by random seeds, as the seeds in the official IDQL code are selected randomly (as noted in line 21 of “hyperparameter.py”). In our code, the seeds are also randomly selected.
>
> For the AntMaze experiments, the difference arises from both the random seeds and the AntMaze version. We used AntMaze-v2, whereas the authors of the original IDQL paper used AntMaze-v0. We chose AntMaze-v2 because our baseline results from SfBC [1] are based on AntMaze-v2.

---

> ### Author Response · Authors · 2024-11-23
>
> > Q6: Hyperparameter Sensitivity and Ablation Study
>
> In the previous version, the hyperparameters we adjusted for different environments were $N$ and $\eta$. For energy-based AlignIQL (i.e., Diffusion-based AlignIQL, abbreviated as D-AlignIQL), the key hyperparameter is $N$, which represents the number of actions generated by the diffusion-based behavior policy. In this setting, $N$ has a greater influence on performance than $\eta$, as a higher $N$ increases the likelihood of finding the “lucky” action that satisfies $\hat{a} = \arg\max_a Q(s, a)$, and $\eta$ is applied across all evaluated actions.
>
> For $N$, Figures 1 and 2 show that the performance of D-AlignIQL improves as $N$ increases, whereas IDQL does not exhibit a similar trend. For $\eta$, we reran the MuJoCo tasks using the policy-based implementation of AlignIQL (abbreviated as AlignIQL) to analyze the impact of $\eta$. Below are the results of the ablation study on $\eta$.
>
> **Performance of AlignIQL Under Different $ \eta $**
>
> | $\eta$ | Walker2d ME | Walker2d MR | Halfcheetah ME | Halfcheetah MR |
> |----------|-------------|-------------|----------------|----------------|
> | $\eta=3$ | 110.3      | 77.4        | 82.1           | 42.6           |
> | $\eta=5$ | 110.4      | 79.5        | 81.4           | 42.7           |
> | $\eta=10$ | 110.5     | 80.1        | 80.1           | 42.5           |
>
> We have added this to the revised paper.
>
> > Q7: The authors state, “Figure 2 shows that as training time increases, the performance of AlignIQL with different N converges to the same value, which shows that AlignIQL is insensitive to N.” This conclusion is not obvious from Figure 2. A clearer approach would be to report the mean and standard deviation of these scores.
>
> Thank you for pointing this out. We acknowledge that this conclusion is not immediately obvious from Figure 2. We have added the quantitative results from Figure 2 into Chapter 6.2 to clarify our findings. What we want to emphasize from Figure 2 is that the performance of AlignIQL improves with increasing $N$, whereas IDQL does not exhibit a similar trend. This phenomenon demonstrates the robustness of our method, as we expect that, for a robust method, the performance with different values of $N$ should not degrade.
>
> We have revised this section to make it clearer and have also included an explanation in the revised Chapter 6.3 on why our method benefits from larger $N$. Thank you again for your valuable suggestions.
>
>
> > Q8: "Policy Alignment" Definition Differs from Its Use in Language Models
>
> Thank you for pointing this out. In fact, Definition 4.1 is the definition of policy alignment. We have emphasized this in the revised paper.
>
>
> > Q9: Duplicate References and Editing Errors
>
> We appreciate that Reviewer pv7F identified the errors of duplicate references and editing issues. We have corrected them. Thank you again for your rectification.
>
>
> **References**
>
> [1] Chen, Huayu, et al. “Offline Reinforcement Learning via High-Fidelity Generative Behavior Modeling.” *arXiv Preprint arXiv:2209.14548*, 2022.

---

> ### Author Response · Authors · 2024-11-26
>
> Dear reviewer,
>
> Thank you once again for investing your valuable time in providing feedback on our paper. Your insightful suggestions have led to significant improvements in our work, and we look forward to possibly receiving more feedback from you. Since the discussion period between the author and reviewer is rapidly approaching its end, we kindly request you to review our responses to ensure that we have addressed all of your concerns. Also, we remain eager to engage in further discussion about any additional questions you may have.
>
> Best,
>
> Authors

---

> ### Author Response · Authors · 2024-12-03
>
> Dear reviewer,
>
> Since the discussion period between the author and reviewer is rapidly approaching its end, we were wondering if our response and revision have resolved your concerns. In our responses, we focus on clarifying the simplicity, generalizability, and performance of the proposed method and provide extensive experiments. If our response has addressed your concerns, we would be grateful if you could re-evaluate our work.
>
> Best regards,
>
> The Authors

---

> > ### Comment · Reviewer_pv7F · 2024-12-03
> >
> > Thank you for your response.
> >
> > The definition of policy alignment in Section 4.1 remains unclear. I recommend providing an explicitly formal definition, as it is a central concept of the paper.
> >
> > The explanation regarding random seeds is acceptable.
> >
> > However, my concerns about the experiments remain unresolved. These include issues related to the increasing computational costs, sensitivity to hyperparameters, and the results from the Mujoco suite.

---

> ### Author Response · Authors · 2024-12-03
>
> Thank you for your feedback. Here, we have once again carefully addressed your concerns.
> > The definition of policy alignment
>
> We formally define policy alignment as the condition where the extracted policy from the learned value function satisfies Definition 4.1. We will update it in the final version. In addition, I believe it is unlikely that policy alignment in our paper would be misunderstood as policy alignment in Language Models.
>
> > Experiments about computational costs
>
> We have added extensive experiments to validate the computational costs (see Appendix F.2 and the response to Reviewer KMNe Q4). In short, the runtime of Diffusion-based-AlignIQL is comparable to other diffusion-based methods. More importantly, our method can be combined with diffusion-based acceleration methods, such as EDP [1], to significantly reduce training time, as demonstrated in Appendix F.2.
>
> > Experiments about MuJoCo
>
> To address your concerns about MuJoCo, we have rerun the MuJoCo experiments using AlignIQL (average score $78.5$) without employing the Diffusion Model to demonstrate that our policy extraction method achieves performance comparable to AWR+IQL (average score $76.9$). (See Table 1 in the revised paper.) We also explain the reasons for the poorer performance of D-AlignIQL in MuJoCo tasks, which may be attributed to the following:
>
> 1. The saturated performance in MuJoCo tasks, where the impact of policy alignment is less pronounced.
>
> 2. The objective function (i.e., the regularizer) in Equation IPF and the diffusion model, which constrain the distance between the learned policy and the behavior policy, result in overly conservative actions.
>
> > Sensitivity to hyperparameters
>
> There are two main hyperparameters in our paper: $\eta$ for AlignIQL (without the diffusion model), and $\eta$ and $N$ for D-AlignIQL (with the diffusion model). The ablation study on $\eta$ for AlignIQL can be found in Appendix F.3, and we also present it here.
> | $\eta$ | Walker2d ME | Walker2d MR | Halfcheetah ME | Halfcheetah MR |
> |----------|-------------|-------------|----------------|----------------|
> | $\eta=3$ | 110.3      | 77.4        | 82.1           | 42.6           |
> | $\eta=5$ | 110.4      | 79.5        | 81.4           | 42.7           |
> | $\eta=10$ | 110.5     | 80.1        | 80.1           | 42.5           |
>
> For D-AlignIQL, we conducted an ablation study on $N$. In this setting, the hyperparameter $N$ has a greater influence on performance than $\eta$, as a higher $N$ increases the likelihood of finding the “lucky” action that satisfies $\hat{a} = \arg\max_a Q(s, a)$. Below are the results of the ablation study for different values of $N$:
>
> | D4RL Tasks | IDQL (N=16) | IDQL (N=64) | IDQL (N=256) | D-AlignIQL (N=16) | D-AlignIQL (N=64) | D-AlignIQL (N=256) |
> |------------|--------------|--------------|---------------|--------------------|--------------------|--------------------|
> | AntMaze    | 72.0         | 66.5         | 58.8          | 65.8              | 70.2              | 70.7              |
>
> As shown in the above table and Section 6.3, the performance of D-AlignIQL improves with increasing $N$, whereas IDQL does not exhibit a similar trend. This phenomenon demonstrates the robustness of our method, as a robust method should maintain or improve its performance across different values of $N$ without degradation.
>
> In summary, we have carefully addressed all the concerns raised by Reviewer pv7F and incorporated the solutions into the revised paper. Due to the delayed response to the reviewer, we kindly request that you carefully review our replies to ensure that we have addressed all the concerns you raised.
>
> [1] Kang, Bingyi, et al. Efficient Diffusion Policies for Offline Reinforcement Learning.

---

### Official Review · Reviewer_g3hi · 2024-11-04

**Soundness:** 3
**Presentation:** 2
**Contribution:** 3
**Rating:** 6
**Confidence:** 3

**Summary:**

The paper considers policy extraction problem, where sometimes in offline RL, existing algorithms only learn value function, and policy extraction problem is to find a policy that coorespond to the policy and does not perform OOD actions. The paper considers distilling policies from value functions learned with IQL algorithm, and propose the implicit policy-finding problem. The solution of the IPF problem leads to the proposed AlignIQL algorithm, from a careful derivation of the IPF formulation. In the experiment, the proposed method is compared with several baselines with competitive performance.

**Strengths:**

1. The proposed method is derived rigorously.
2. The experiment shows that the proposed method has good empirical performance compared with other baselines on standard benchmarks.

**Weaknesses:**

1. The formulation aims to use a general regularization function $f$, which is a good attempt. However, the remaining results seems to rely on the case that $f(x) = \log(x)$. Does the result generalize to any other regularization function?
2. Remark 5.7 seems very hand-wavy. How does the algorithm ensure that the action with the positive advantage is chosen? It does not seem to be reflected in the loss function.
3. While the result in table 1 looks impressive, I am not sure if this can serve a strong evidence that the proposed method is better than AWR. The proposed method is equipped with diffusion policies, but the IQL (AWR) baseline seem to only use MLP so it might not be a fair comparison.
4. The result in table 1 is missing standard deviation.
5. The goal of section 6.2 is unclear. What is the baseline that is compared against in this section?
6. Some minor issues: a) in eq. 1, is a $\pi(a \mid s)$ missing? b) in eq. 2, where is the $Q_{\theta}$ from? It does not appear eq. 1. c) in eq. 7, the notation $a$ is overloaded in $a' \sim \pi(a \mid s)$.

**Questions:**

see above

---

> ### Author Response · Authors · 2024-11-23
>
> We thank reviewer g3hi for the comments and suggestions. Below, we address the concerns raised in your review point by point.
>
> Q1 Other regularization function
>
> We have conducted the experiment of linear regularization function $ f(x)=x-1 $ in Appendix H. Here are the results
>
> | Regularizers | AlignIQL (umaze-p) | AlignIQL (umaze-d) | AlignIQL-hard (umaze-p) | AlignIQL-hard (umaze-d) |
> | --- | --- | --- | --- | --- |
> |  $ f(x) = \log x $  | 94.8 | 82.4 | 84.7 | 74.0 |
> | $ f(x) = x-1 $  | 95.0 | 87.0 | 92.0 | 70.0 |
>
>
> We found that the performance of the linear regularizer is comparable to the results of AlignIQL. This is because both place more weight on actions with higher $ \{Q(s,a)-V(s)\}^2 $. For AlignQIL-hard, using a linear regularizer can enhance performance in certain cases, as it prevents numerical explosion caused by the exponential function.
>
> > Q2 Remark 5.7 seems very hand-wavy. How does the algorithm ensure that the action with the positive advantage is chosen? It does not seem to be reflected in the loss function.
>
>
> There are two methods to use our method.
>
> 1. Energy-based implementation: We first use the learned diffusion-based behavior model $ \mu_\phi(a|s) $ to generate $N$ action samples. These actions are then evaluated using weights from AlignIQL(Eq 15) In this setting, we do not need the loss function and select actions according to the weights from AlignIQL(Eq 15).
> 2. Policy-based implementation: We use Eq 16 or Eq 12 to train the policy, which needs the exact probability density of the current policy.  In this setting, the loss function is weighed by $-\eta(Q-V)^2$
>
> Next, we will explain why $ -\eta(Q-V)^2,\eta>0 $ can choose the action with optimal value function.
>
> For IQL and $ \eta>0 $, the expectile loss approximates the maximum of    $ Q_{\hat{\theta}}(s,a) $ when $ \tau\approx 1 $.  We can approximately think $V(s)=\arg\max_{a\sim \mathcal{D}} Q(s,a)$, and thus, according to Eq 15, $ \hat{a} = \arg\max_a Q(s, a) $ has a weight of 1, while other actions are weighted by $ \exp{{-\eta (Q(s, a) - V(s))^2}} $. For a fixed $ \eta $, the weights for other actions are smaller than $\arg\max_{a\sim \mathcal{D}} Q(s,a)$. Therefore, Eq 15 approximately recovers the implicit policy $\pi^*(a|s)=\arg\max_{a\sim\mathcal{D}} Q^*(s,a)$ from IQL learned value functions.
>
> We treated AWR as a special case of AlignIQL (if $ \eta=-1, Q > V $ and sometimes referred to AWR as AlignIQL with $ \eta=-1 $. This could be considered imprecise. Therefore, we reran our experiments on the D4RL datasets. Now, all experiments for AlignIQL are conducted with positive $ \eta $ as shown above, where such $\eta$ can recover the optimal policy $ \pi^{}(a|s) = \arg\max_{a \sim \mathcal{D}} Q^*(s, a) $ under the optimal value functions.
>
>
>
> We have added this to our revised paper for clarity.
>
>
> >Q3 Compared to AWR
>
>
> In fact, SfBC [1] is a diffusion-based method that selects actions according to AWR. AlignIQL outperforms it in 5 out of 6 AntMaze tasks. The score of SfBC is reported from its original paper. We also compared the Diffusion model + AWR in Section 6.2. Additionally, we conducted experiments on different regularizers, vision-based discrete control, and robustness to demonstrate the generalizability of our method, as presented in Appendix H–J.
>
>
>
> > Q4 missing standard deviation
>
> Thank you for pointing this out. We previously ignored the standard deviation, as IDQL does not include it. We have now added the standard deviation to our D4RL results.
>
> > Q5 some typos
>
> We appreciate that Reviewer g3hi identified our typos. We have corrected them and thank you again for your rectification.
>
> [1] Chen, Huayu, et al. “Offline Reinforcement Learning via High-Fidelity Generative Behavior Modeling.” _arXiv Preprint arXiv:2209.14548_, 2022.

---

> ### Author Response · Authors · 2024-11-26
>
> Dear reviewer,
>
> Thank you once again for investing your valuable time in providing feedback on our paper. Your insightful suggestions have led to significant improvements in our work, and we look forward to possibly receiving more feedback from you. Since the discussion period between the author and reviewer is rapidly approaching its end, we kindly request you to review our responses to ensure that we have addressed all of your concerns. Also, we remain eager to engage in further discussion about any additional questions you may have.
>
> Best,
>
> Authors

---

> > ### Comment · Reviewer_g3hi · 2024-12-02
> >
> > I appreciate the authors for their detailed rebuttal. Quick followup: my first concern on the regularization function is mostly on the theoretical side: does the derivation of the objective still hold if we use anything other than $f(x) = \log(x)$?

---

> ### Author Response · Authors · 2024-12-03
>
> Thank you for your response. Our method can be extended to any regularization function as long as the function satisfies Assumption 4.3. In fact, we have already derived the general form of $f(x)$ under Assumption 4.3 in Equations 9 and 14. We also conducted experiments on $f(x) = x - 1$ in Appendix H. Detailed derivations can also be found in Appendix A and H.
>
> The results of different regularizers are as follows:
> | Regularizers     | D-AlignIQL (umaze-p) | D-AlignIQL (umaze-d) | D-AlignIQL-hard (umaze-p) | D-AlignIQL-hard (umaze-d) |
> |------------------|-----------------------|-----------------------|---------------------------|---------------------------|
> |  $f(x) = \log x$ | 94.8                 | 82.4                 | 84.7                     | 74.0                     |
> | $f(x) = x - 1$   | 95.0                 | 87.0                 | 92.0                     | 70.0                     |
>
>  Since the discussion period between the author and reviewer is rapidly approaching its end, we kindly request you to review our responses to ensure that we have addressed all of your concerns. Also, we remain eager to engage in further discussion about any additional questions you may have.

---

> > ### Comment · Reviewer_g3hi · 2024-12-03
> >
> > Thank you for your response! I think my concerns are addressed and I have increased my score.

---

> > > ### Author Response · Authors · 2024-12-03
> > >
> > > Thanks for increasing the score! Your constructive feedback has been instrumental in improving the quality of our work and we deeply appreciate your willingness to increase the score.

---

### Author Response · Authors · 2024-11-23
**Response to all**

We thank the reviewers for their valuable feedback and appreciate the great efforts made by all reviewers, ACs, SACs, and PCs.

We have revised our paper in response to the feedback and summarize the main updates below for convenience:

1. We have added extensive experiments on vision-based discrete control, robustness, and different regularizers. (**Appendix H I J**)

2. We have rerun the MuJoCo experiments to demonstrate that our method is easily integrated with IQL. (**Table 1, Section 6**)

3. We have revised the experimental and method sections to highlight our contributions and reduce ambiguities. (**Section 5,6**)

4. We have updated the hyperparameter table and definition of policy alignment to improve clarity. (**Appendix F, Section 4** )

5. We have added an ablation study on $\eta$. (**Appendix F.3**)

6. We have corrected typos, removed duplicate references, and included standard deviations in the main D4RL results. (**Section 6**)

7. We have conducted extensive experiments to demonstrate that the computational cost of our method is acceptable and that **it is highly easy to implement**. (**Appendix F.2**)

---

### Meta-Review · Area_Chair_ymjN · 2024-12-21

**Metareview:**

The paper studied how to distill policies from value functions learned with the IQL algorithm and proposed the implicit policy-finding problem. The author formulated this as a constrained optimization problem and derived two versions of alignIQL. Experiment results show that AlignIQL is compatible with diffusion model based policies and achieved competitive performance under standard offline RL benchmarks. The weaknesses of the paper are in the experiment,

**Additional Comments On Reviewer Discussion:**

The reviewers all acknowledged the authors's efforts during the AC-reviewer discussion phase. However, reviewers were not fully convinced by the additional experiment results provided by the authors. One reviewer still thinks the results are insignificant compared to baselines IQL/CQL. The authors should also try more random seeds to more faithfully replicate the results from one prior work more faithfully. One reviewer is still concerned about the additional complexity, computation cost, and hyperparameter sensitivity introduced by the algorithm, and is unconvinced by the Mujoco results. We encourage the authors to carefully include the newly conducted experiments in future versions of the paper, and we believe this can make the paper much stronger.

---

### Decision · Program_Chairs · 2025-01-22

Reject